# Diffusion Probabilistic Models for Structured Node Classification

**Hyosoon Jang**[1], **Seonghyun Park**[1], **Sangwoo Mo**[2], **Sungsoo Ahn**[1]
[1]POSTECH    [2]University of Michigan
{hsjang1205,shpark26,sungsoo.ahn}@postech.ac.kr, swmo@umich.edu

## Abstract

This paper studies structured node classification on graphs, where the predictions should consider dependencies between the node labels. In particular, we focus on solving the problem for partially labeled graphs where it is essential to incorporate the information in the known label for predicting the unknown labels. To address this issue, we propose a novel framework leveraging the diffusion probabilistic model for structured node classification (DPM-SNC). At the heart of our framework is the extraordinary capability of DPM-SNC to (a) learn a joint distribution over the labels with an expressive reverse diffusion process and (b) make predictions conditioned on the known labels utilizing manifold-constrained sampling. Since the DPMs lack training algorithms for partially labeled data, we design a novel training algorithm to apply DPMs, maximizing a new variational lower bound. We also theoretically analyze how DPMs benefit node classification by enhancing the expressive power of GNNs based on proposing AGG-WL, which is strictly more powerful than the classic 1-WL test. We extensively verify the superiority of our DPM-SNC in diverse scenarios, which include not only the transductive setting on partially labeled graphs but also the inductive setting and unlabeled graphs.

## 1  Introduction

In this paper, we address the node classification problem, which is a fundamental problem in machine learning on graphs with various applications, such as social networks [1] and citation networks [2]. One representative example is a transductive problem to classify documents from a partially labeled citation graph. Recently, graph neural networks (GNNs) [3, 4] have shown great success in this problem over their predecessors [5, 6]. Their success stems from the high capacity of non-linear neural architectures combined with message passing.

However, the conventional GNNs are incapable of *structured node classification*: predicting node labels while considering the node-wise label dependencies [7–9]. That is, given a graph $G$ with vertices $\mathcal{V}$ and node labels $\{y_i : i \in \mathcal{V}\}$, a GNN with parameter $\theta$ outputs an unstructured prediction, i.e., $p_\theta(y_i, y_j | G) = p_\theta(y_i | G) p_\theta(y_j | G)$ for $i, j \in \mathcal{V}$. Especially, this limitation becomes problematic in a transductive setting where the prediction can be improved by incorporating the known labels, e.g., output $p_\theta(y_i | G, y_j)$ for known $y_j$. In Figures 1(a) and 1(b), we elaborate on this issue with an example where the conventional GNN fails to make the correct prediction.

To resolve this issue, recent studies have investigated combining GNNs with classical structured prediction algorithms, i.e., schemes that consider dependencies between the node labels [8–13]. They combine GNNs with conditional random fields [14], label propagation [15], or iterative classification algorithm [2]. Despite the promising outcomes demonstrated by these studies, their approach relies on the classical algorithms to express joint dependencies between node labels and may lack sufficient expressive power or consistency in incorporating known labels.

37th Conference on Neural Information Processing Systems (NeurIPS 2023).

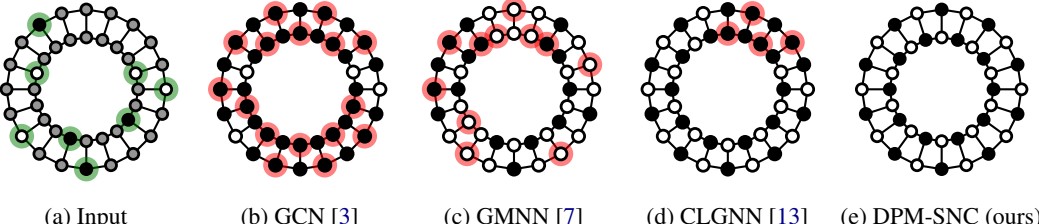

(a) Input      (b) GCN [3]      (c) GMNN [7]      (d) CLGNN [13]      (e) DPM-SNC (ours)

Figure 1: Results of various methods to solve node classification on a non-attributed and partially labeled cyclic grid. (a) The task is to estimate $p(\boldsymbol{y}_U | \boldsymbol{y}_L, G)$ where the known labels $\boldsymbol{y}_L$ are highlighted in green and the unknown labels $\boldsymbol{y}_U$ are colored in gray. (b)-(e) Red highlights the incorrect predictions made by the corresponding algorithm. Conventional GNN (GCN) fails to make an informative prediction while our DPM-SNC is the only method to perfectly predict all the labels.

**Contribution.** We propose a novel framework for structured node classification called DPM-SNC. Our key idea is to leverage the diffusion probabilistic model (DPM) [16], motivated by two significant advantages of DPM for structured node classification: (a) it can effectively learn a joint distribution over both the unknown and the known labels, and (b) it can easily incorporate conditions during inference via posterior sampling. Figure 1 highlights that DPM significantly outperforms previous methods when the prediction problem requires a complete understanding of label dependencies.

However, DPM cannot be directly applied to the transductive scenario, where the model needs to maximize its log-likelihood for partially labeled graphs. We propose a novel training algorithm to address this challenge. Our method maximizes a new variational lower bound of the marginal likelihood of the graph over the unlabeled nodes, involving the alternative optimization of DPM and a variational distribution. In this process, we estimate the variational distribution from the current DPM using a similar approach to fixed-point iteration [17]. After training, DPM can predict the remaining node labels by constraining its predictions with the given labeled nodes, utilizing the manifold-constrained sampling technique [18].

In addition, we provide a theoretical analysis that supports the benefits of DPM-SNC for node classification by enhancing the expressive power of GNNs. To this end, we derive an analog of our DPM-SNC based on the Weisfeiler-Lehman (WL) [19] test, evaluating the ability to distinguish non-isomorphic graphs. Specifically, we prove that DPM-SNC is as powerful as our newly derived aggregated Weisfeiler-Lehman (AGG-WL) test, which is strictly more powerful than the 1-WL test, a known analog to standard GNN architectures [20] such as GCN [3] and GAT [21].

We demonstrate the effectiveness of DPM-SNC on various datasets, covering both transductive and inductive settings. In the transductive setting, we conduct experiments on seven benchmarks: Pubmed, Cora, and Citeseer [22]; Photo and Computer [23]; and Empire and Ratings [24]. Additionally, we introduce a synthetic benchmark to evaluate the ability to capture both short and long-range dependencies. DPM-SNC outperforms the baselines, including both homophilic and heterophilic graphs. In the inductive setting, we conduct experiments on four benchmarks: Pubmed, Cora, Citeseer, and PPI [25]. Furthermore, we evaluate DPM-SNC on the algorithmic reasoning benchmark [26], which requires a higher-level understanding of node relations. DPM-SNC also outperforms previous algorithms for these tasks.

To summarize, our contribution can be listed as follows:

- We explore DPM as an effective solution for structured node classification due to its inherent capability in (a) learning the joint node-wise label dependency in data and (b) making predictions conditioned on partially known data (Section 3).

- We propose a novel method for training a DPM on partially labeled graphs that leverages the probabilistic formulation of DPM to maximize the variational lower bound (Section 4).

- We provide a theoretical analysis of how DPM enhances the expressive power of graph neural networks, strictly improving its capability over the 1-WL test (Section 5).

- Our experiments demonstrate the superiority over the baselines in both transductive and inductive settings. We additionally consider heterophilic graphs and algorithmic reasoning benchmarks to extensively evaluate the capability of DPMs in understanding the label dependencies (Section 6).

## 2 Related Work

**Graph neural networks (GNNs).** Graph neural networks (GNNs) are neural networks specifically designed to handle graph-structured data. Over the past years, GNNs have shown promising outcomes in various tasks related to graph representation learning [3, 4, 27, 28]. In the case of node classification, GNNs aggregate neighboring node information and generate node representations for label predictions. Their expressive power has been analyzed through the lens of the Weisfeiler-Lehman (WL) test [19], a classical algorithm to distinguish non-isomorphic graphs. For example, Xu et al. [20] proved how the expressive power of conventional GNNs is bounded by the 1-dimensional WL (1-WL) test.

**Structured node classification with GNNs.** Apart from investigating the power of GNNs to discriminate isomorphic graphs, several studies focused on improving the ability of GNNs to learn the joint dependencies [7, 9, 13, 26, 29–31]. In particular, researchers have considered GNN-based structured node classification [7, 9, 13]. First, Qu et al., [7] parameterize the potential functions of conditional random fields using a GNN and train it using pseudo-likelihood maximization [32]. Next, Qu et al., [9] proposed a training scheme with a GNN-based proxy for the conditional random field. Hang et al., [13] proposed an iterative classification algorithm-based learning framework of GNNs that updates each node-wise prediction by aggregating previous predictions on neighboring nodes.

**Diffusion probabilistic models (DPMs).** Inspired by non-equilibrium thermodynamics, DPMs are latent variable models (LVMs) that learn to reverse a diffusion process and construct data from a noise distribution. Recent studies demonstrated great success of DPMs in various domains, e.g., generating images [33], text [34], and molecules [35]. An important advantage of diffusion models is their ability to incorporate additional constraints. This allows DPM to solve the posterior inference problems, e.g., conditioning on the partially observed data to recover the original data [18, 36, 37].

## 3 Diffusion Probabilistic Models for Structured Node Classification

In this section, we introduce the problem of structured node classification and discuss the limitations of graph neural networks (GNNs) in addressing this problem (Section 3.1). We then explain how the diffusion probabilistic models (DPMs) offer a promising solution for this problem (Section 3.2).

### 3.1 Structured node classification

We address structured node classification, which involves predicting node labels while considering their joint dependencies. Specifically, we focus on predicting labels in partially labeled graphs, where some true labels are known in advance.[1] Our goal is to devise an algorithm that considers the dependencies with known labels to enhance predictions for unknown labels.

To be specific, our problem involves a graph $G = (\mathcal{V}, \mathcal{E}, \boldsymbol{x})$ consisting of nodes $\mathcal{V}$, edges $\mathcal{E}$, and node attributes $\boldsymbol{x} = \{x_i : i \in \mathcal{V}\}$. We also denote the node labels by $\boldsymbol{y} = \{y_i : i \in \mathcal{V}\}$. We let $\mathcal{V}_L$ and $\mathcal{V}_U$ denote the set of labeled and unlabeled nodes, while $\boldsymbol{y}_L$ and $\boldsymbol{y}_U$ denote the corresponding labels for each set, e.g., $\boldsymbol{y}_L = \{y_i : i \in \mathcal{V}_L\}$. Our objective is to predict the unknown labels $\boldsymbol{y}_U$ by training on the partially labeled graph, aiming to infer the true conditional distribution $p(\boldsymbol{y}_U|G, \boldsymbol{y}_L)$.

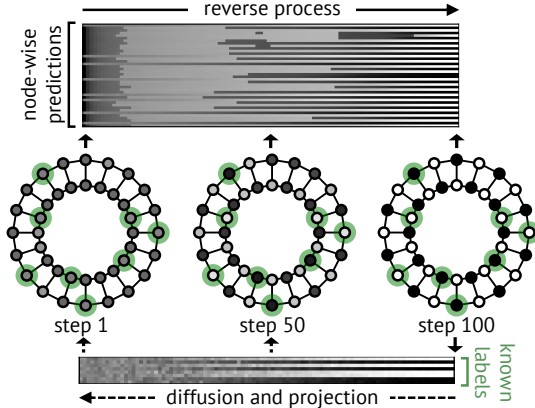

Figure 2: The reverse process makes predictions with label dependencies. The node color indicates the label value, and opacity reflects the likelihood.

We point out that the conventional GNNs [3, 4, 27, 28] are suboptimal for estimating $p(\boldsymbol{y}_U|G, \boldsymbol{y}_L)$ since their predictions on the node labels are independently conditioned on the input, i.e., their predictions are factorized by $p_\theta(\boldsymbol{y}_U|G, \boldsymbol{y}_L) = \prod_{i \in \mathcal{V}_U} p_\theta(y_i|G)$. Namely, conventional GNNs cannot incorporate the information of the known labels $\boldsymbol{y}_L$ into a prediction of the unknown labels $\boldsymbol{y}_U$.

---

[1]We also show structured node classification to be important for unlabeled graphs in Section 5.

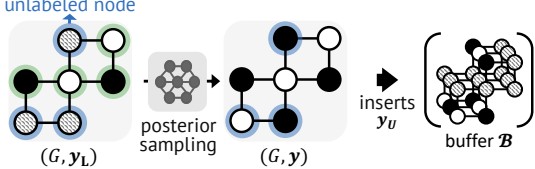

(a) Update the buffer with samples from $p_{\boldsymbol{\theta}}(\boldsymbol{y}_U | G, \boldsymbol{y}_L)$.

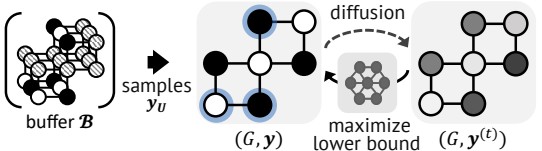

(b) Update the DPM parameter to maximize $\mathcal{L}_{\text{VLB}}$.

Figure 3: Illustration of DPM-SNC training on partially labeled graphs, repeating steps (a) and (b).

**Algorithm 1** DPM-SNC
1: Train a mean-field GNN $p_{\phi}(\boldsymbol{y}|G)$.
2: Initialize the buffer $\mathcal{B}$ by $p_{\phi}(\boldsymbol{y}_U | G)$.
3: **repeat**
4:  **for** $i = 1, \ldots, N_1$ **do**
5:    Get $\boldsymbol{y}_U \sim p_{\boldsymbol{\theta}}(\boldsymbol{y}_U | G, \boldsymbol{y}_L)$ using manifold-constrained sampling.
6:    Update $\mathcal{B} \leftarrow \mathcal{B} \cup \{\boldsymbol{y}_U\}$.
7:    If $|\mathcal{B}| > K$, remove oldest one in $\mathcal{B}$.
8:  **end for**
9:  **for** $i = 1, \ldots, N_2$ **do**
10:   Sample $\boldsymbol{y}_U \sim \mathcal{B}$.
11:   Update $\theta$ to maximize $\mathcal{L}_{\text{VLB}}$ with $G$, and $\boldsymbol{y} = \boldsymbol{y}_L \cup \boldsymbol{y}_U$.
12:  **end for**
13: **until** converged

Structured prediction algorithms [2, 14] overcome this issue by solving two tasks: (a) learning a joint distribution $p_{\theta}(\boldsymbol{y}_U, \boldsymbol{y}_L | G)$ to maximize the likelihood of labeled data $p_{\theta}(\boldsymbol{y}_L | G)$ and (b) inferring from the distribution $p_{\theta}(\boldsymbol{y}_U | G, \boldsymbol{y}_L)$ conditioned on known labels.

## 3.2 Diffusion probabilistic models for structured node classification

In this work, we consider diffusion probabilistic models for structured node classification (DPM-SNC) motivated by how their strengths align with the essential tasks for solving structured node classification, outlined in Section 3.1. In particular, DPMs are equipped with (a) high expressivity in learning a joint distribution over data and (b) the ability to easily infer from a posterior distribution conditioned on partially observed data.

To this end, we formally describe DPMs that generate node-wise labels $\boldsymbol{y}$ associated with a graph $G$. At a high level, our DPM consists of two parts: the forward process, which injects noise into the labels, and the reverse process, which denoises to predict the true labels $\boldsymbol{y}$ [16]. Given the number of diffusion steps $T$, the forward process constructs a sequence of noisy labels $\boldsymbol{y}^{(1:T)} = [\boldsymbol{y}^{(1)}, \ldots, \boldsymbol{y}^{(T)}]$ using a fixed distribution $q(\boldsymbol{y}^{(1:T)} | \boldsymbol{y}^{(0)})$ starting from the true label $\boldsymbol{y}^{(0)} = \boldsymbol{y}$.[2] Next, given an initial noises $\boldsymbol{y}^{(T)}$ sampled from $p(\boldsymbol{y}^{(T)})$, the reverse process $p_{\boldsymbol{\theta}}(\boldsymbol{y}^{(0:T-1)} | \boldsymbol{y}^{(T)}, G)$ is trained to recover the forward process conditioned on the graph $G$. We defer the detailed parameterization to Section 4. To be specific, the forward and the reverse process are factorized as follows:

$$q(\boldsymbol{y}^{(1:T)} | \boldsymbol{y}^{(0)}) = \prod_{t=1}^{T} q(\boldsymbol{y}^{(t)} | \boldsymbol{y}^{(t-1)}), \qquad p_{\boldsymbol{\theta}}(\boldsymbol{y}^{(0:T-1)} | \boldsymbol{y}^{(T)}, G) = \prod_{t=1}^{T} p_{\boldsymbol{\theta}}(\boldsymbol{y}^{(t-1)} | \boldsymbol{y}^{(t)}, G),$$

where the training of the reverse process is trivial on a fully-labeled graph $(\boldsymbol{y}, G)$ by maximizing the variational lower bound of the marginal log-likelihood $\log p_{\boldsymbol{\theta}}(\boldsymbol{y}|G) = \log \sum_{\boldsymbol{y}^{(1:T)}} p_{\boldsymbol{\theta}}(\boldsymbol{y}^{(0:T)} | G)$ with respect to the noisy labels $\boldsymbol{y}^{(1:T)}$ sampled from the forward process [16].

By leveraging shared latent variables $\boldsymbol{y}^{(t)}$ across multiple steps in the reverse process, the reverse process effectively considers the dependencies in the output. Next, DPMs can easily infer from a posterior distribution conditioned on partially observed data. Specifically, the incremental updating of $\boldsymbol{y}^{(t)}$ for $t = 1, \ldots, T$ allows the DPM to incorporate the known label $\boldsymbol{y}_L$, e.g., applying iterative projection with manifold-based correction [18]. This enables inference from the distribution $p_{\boldsymbol{\theta}}(\boldsymbol{y}_U | G, \boldsymbol{y}_L)$ conditioned on the known labels $\boldsymbol{y}_L$. See Figure 2 for an illustration of DPM for structured node classification with known labels.

---

[2]We relax the discrete labels to one-hot vectors to apply the forward process.

# 4 Training Diffusion Probabilistic Models on Partially Labeled Graphs

Despite the potential of DPMs for structured node classification, they lack the algorithms to learn from partially labeled graphs, i.e., maximize the likelihood of known labels. To resolve this issue, we introduce a novel training algorithm for DPM-SNC, based on maximizing a variational lower bound for the log-likelihood of known labels.

**Variational lower bound.** At a high-level, our algorithm trains the DPM to maximize the log-likelihood of training data $\log p_\theta(\boldsymbol{y}_L|G)$, which is defined as follows:

$$\mathcal{L} = \log p_\theta(\boldsymbol{y}_L|G) = \log \sum_{\boldsymbol{y}_U} \sum_{\boldsymbol{y}^{(1:T)}} p_\theta(\boldsymbol{y}_L, \boldsymbol{y}_U, \boldsymbol{y}^{(1:T)}|G).$$

However, this likelihood is intractable due to the exponentially large number of possible combinations for the unknown labels $\boldsymbol{y}_U$ and the noisy sequence of labels $\boldsymbol{y}^{(1:T)}$. To address this issue, we train the DPM based on a new variational lower bound $\mathcal{L} \geq \mathcal{L}_{\text{VLB}}$, expressed as follows:

$$\mathcal{L}_{\text{VLB}} = \mathbb{E}_{q(\boldsymbol{y}_U|\boldsymbol{y}_L)}\left[\mathbb{E}_{q(\boldsymbol{y}^{(1:T)}|\boldsymbol{y})}\left[\log p_\theta(\boldsymbol{y}, \boldsymbol{y}^{(1:T)}|G) - \log q(\boldsymbol{y}^{(1:T)}|\boldsymbol{y})\right] - \log q(\boldsymbol{y}_U|\boldsymbol{y}_L)\right]. \quad (1)$$

Here, $q(\cdot)$ is a variational distribution factorized by $q(\boldsymbol{y}_U, \boldsymbol{y}^{(1:T)}|\boldsymbol{y}_L) = q(\boldsymbol{y}^{(1:T)}|\boldsymbol{y})q(\boldsymbol{y}_U|\boldsymbol{y}_L)$, where $\boldsymbol{y} = \boldsymbol{y}_U \cup \boldsymbol{y}_L$. We provide detailed derivation in Appendix A.1.

**Parameterization.** We define the reverse process $p_\theta(\boldsymbol{y}^{(t-1)}|\boldsymbol{y}^{(t)}, G)$ with a Gaussian distribution, where the mean value parameterized by a graph neural network (GNN) and the variance set to be a hyper-parameter. Following the prior work [16], we also employ a Gaussian diffusion to parameterize the forward process $q(\boldsymbol{y}^{(1:T)}|\boldsymbol{y})$ as follows:

$$q(\boldsymbol{y}^{(1)}, \ldots, \boldsymbol{y}^{(T)}|\boldsymbol{y}^{(0)}) = \prod_{t=1}^{T} \mathcal{N}(\boldsymbol{y}^{(t)}; \sqrt{1-\beta_t}\boldsymbol{y}^{(t-1)}, \beta_t\mathbf{I}),$$

where $\mathbf{I}$ is an identity matrix, $\beta_1, \ldots, \beta_T$ are fixed variance schedules. We set the variance schedule to promote $q(\boldsymbol{y}^{(T)}|\boldsymbol{y}^{(0)}) \approx \mathcal{N}(\boldsymbol{y}^{(T)}; \mathbf{0}, \mathbf{I})$ by setting $\beta_t < \beta_{t+1}$ for $t = 0, \ldots, T-1$ and $\beta_T = 1$. Finally, we describe the variational distribution $q(\boldsymbol{y}_U|\boldsymbol{y}_L)$ as an empirical distribution over a fixed-size buffer $\mathcal{B}$ containing multiple estimates of the unknown labels $\boldsymbol{y}_U$. This buffer is updated throughout the training. We provide more details on our parameterization in Appendix A.2.

**Training algorithm.** To maximize the variational lower bound, we alternatively update the GNN-based reverse process $p_\theta(\boldsymbol{y}, \boldsymbol{y}^{(1:T)}|G)$ and the buffer-based variational distribution $q(\boldsymbol{y}_U|\boldsymbol{y}_L)$. In particular, we update the parameters $\theta$ of the reverse process $p_\theta(\boldsymbol{y}, \boldsymbol{y}^{(1:T)}|G)$ to maximize the Monte Carlo approximation of $\mathcal{L}_{\text{LVB}}$ by applying ancestral sampling to the variational distribution $q(\boldsymbol{y}^{(1:T)}|\boldsymbol{y})q(\boldsymbol{y}_U|\boldsymbol{y}_L)$, i.e., sampling $\boldsymbol{y}_U$ from the buffer $\mathcal{B}$ and applying the diffusion process to $\boldsymbol{y}$. This is the same as the original denoising-based training of DPM [16] with estimated unknown labels, where the detailed training objective is described in Appendix A.3.

Next, we update the variational distribution $q(\boldsymbol{y}_U|\boldsymbol{y}_L)$ by inserting samples from the distribution $p_\theta(\boldsymbol{y}_U|G, \boldsymbol{y}_L)$ into the buffer $\mathcal{B}$.[3] This update is derived from the condition $q(\boldsymbol{y}_U|\boldsymbol{y}_L) = p_\theta(\boldsymbol{y}_U|G, \boldsymbol{y}_L)$ being necessary to maximize $\mathcal{L}_{\text{VLB}}$, similar to the derivation of fixed-point iteration for optimization [17]. We describe the overall optimization procedure in Figure 3 and Algorithm 1.[4]

Finally, we emphasize that our training algorithm is specialized for DPMs. Previous studies introducing variational lower bounds for structured node classifications [7, 38] face intractability in maximizing $\log p_\theta(\boldsymbol{y}|G)$ or sampling from $p_\theta(\boldsymbol{y}_U|G, \boldsymbol{y}_L)$. They require defining the pseudo-likelihood for the former [32], or parameterizing variational distribution for the latter. However, in our case, the formal simply requires maximizing the lower bound $\mathcal{L}_{\text{VLB}}$, and the latter is easily solved by manifold-constrained sampling [18].

---

[3]We use manifold-constrained sampling of DPM to infer from the distribution $p_\theta(\boldsymbol{y}_U|G, \boldsymbol{y}_L)$ [18]. The detailed sampling procedure is described in Appendix A.4.

[4]In practice, we initialize the buffer $\mathcal{B}$ with samples from a mean-field GNN $p_\phi(\boldsymbol{y}|G)$, which outputs an independent joint distribution over node labels, i.e., $p_\theta(\boldsymbol{y}_U|G, \boldsymbol{y}_L) = \prod_{i \in \mathcal{V}_U} p_\theta(\boldsymbol{y}_i|G)$.

# 5 Theoretical Analysis

In this section, we present a secondary motivation for using DPMs in node classification, distinct from the one given in Section 3 and Section 4. Specifically, we demonstrate that DPMs provably enhance the expressive power of conventional GNNs for solving the graph isomorphism test, implying improved expressive power for node classification problems as well.

To this end, we assess the expressive power of our DPM-SNC by its analog for discriminating isomorphic graphs, which we call the aggregated Weisfeiler-Lehman (AGG-WL) test. Then, we show that AGG-WL is strictly more powerful than the 1-dimensional WL (1-WL) test [19], which is an analog of GNNs [20]. We formalize our results in the following theorem.

**Theorem 1.** *Let 1-WL-GNN be a GNN as powerful as the 1-WL test. Then, DPM-SNC using a 1-WL-GNN is strictly more powerful than the 1-WL-GNN in distinguishing non-isomorphic graphs.*

We provide the formal proof in Appendix B. At a high level, both the 1-WL test and the AGG-WL test assign colors to nodes and iteratively refine them based on the colors of neighboring nodes, enabling the comparison of graph structures. The key difference is that AGG-WL applies perturbations into the initial graph to create multiple augmented graphs, and aggregates the results of the 1-WL algorithm on each augmented graph. The perturbations and aggregation can be done by node-wise concatenations with binary random features and hashing a multiset of the refined graph colors, respectively. Hence, any graph pair distinguishable by the 1-WL test is also distinguishable by our AGG-WL test, and there exist pairs of graphs indistinguishable by the WL test but distinguishable by our variant.

We remark that our analysis can be easily extended to applying other latent variable models, e.g., variational auto-encoders [39] or normalizing flows [40], for node classification. In this analysis, sampling a latent variable corresponds to fixing a perturbation for color assignments at initialization, and aggregating over the perturbed graphs corresponds to computing the prediction over the true labels with marginalization over the latent variables. Our theory is based on that of Bevilaqua et al. [41] which was originally developed for analyzing a particular GNN architecture; our key contribution lies in extending this analysis to the latent variable models.

# 6 Experiments

## 6.1 Transductive setting

We first evaluate the performance of our algorithm in the transductive setting. Specifically, we conduct an evaluation of DPM-SNC on synthetic data, as well as real-world node classification problems. We consider different types of label dependencies by considering both homophilic and heterophilic graphs. We provide the details of our implementation in Appendix C.1.

**Synthetic data.** In this experiment, we create a $2 \times n$ non-attributed cyclic grid. Each node is assigned either a binary label, with neighboring nodes having different labels. We consider two scenarios: one where the known labels are randomly scattered throughout the graph, and another where they are clustered in a local region. Examples of both scenarios are illustrated in Figures 4(a) and 4(b). These two scenarios verify the capability for capturing both short and long-range dependencies between node labels. The data statistics are described in Appendix D.

We compare DPM-SNC with conventional GNN that ignores the label dependencies and other structured node classification methods: GMNN [7], G³NN [8], and CLGNN [13]. We describe the detailed experimental setup in Appendix E.1. We report the performance using five different random seeds.

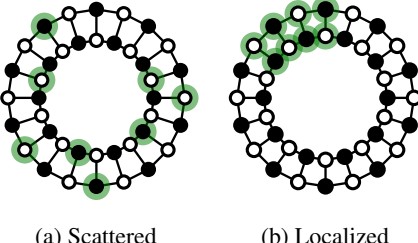

(a) Scattered      (b) Localized

Figure 4: Illustration of two scenarios. Green highlights known labels.

Table 1: The transductive node classification accuracy on synthetic data. **Bold** numbers indicate the best performance.

| Method | Scattered | Localized |
|---|---|---|
| GCN [3] | $50.1_{\pm 0.6}$ | $51.3_{\pm 1.4}$ |
| + GMNN [7] | $64.2_{\pm 3.2}$ | $48.2_{\pm 4.8}$ |
| + G³NN [8] | $50.1_{\pm 0.6}$ | $51.3_{\pm 1.4}$ |
| + CLGNN [13] | $87.0_{\pm 2.4}$ | $53.4_{\pm 2.1}$ |
| + DPM-SNC | $\mathbf{98.5}_{\pm 0.7}$ | $\mathbf{90.9}_{\pm 3.4}$ |

Table 2: The transductive node classification performance. N-Acc. and Sub-Acc. denote the node-level and subgraph-level accuracy, respectively. **Bold** numbers indicate the best performance among the structured prediction methods using the same GNN.

| Method | Pubmed | | Cora | | Citeseer | | Photo | | Computer | |
|---|---|---|---|---|---|---|---|---|---|---|
| | N-Acc. | Sub-Acc. | N-Acc. | Sub-Acc. | N-Acc. | Sub-Acc. | N-Acc. | Sub-Acc. | N-Acc. | Sub-Acc. |
| LP [42] | $69.1_{\pm0.0}$ | $45.7_{\pm0.0}$ | $68.1_{\pm0.0}$ | $46.9_{\pm0.0}$ | $46.1_{\pm0.0}$ | $29.8_{\pm0.0}$ | $81.0_{\pm2.0}$ | $37.2_{\pm1.7}$ | $69.9_{\pm2.9}$ | $15.1_{\pm1.1}$ |
| C&S [43] | $77.3_{\pm0.0}$ | $57.1_{\pm0.0}$ | $80.2_{\pm0.0}$ | $60.1_{\pm0.0}$ | $69.5_{\pm0.0}$ | $47.9_{\pm0.0}$ | $90.1_{\pm1.4}$ | $48.5_{\pm2.5}$ | $81.6_{\pm0.9}$ | $24.5_{\pm1.3}$ |
| BPN [44] | $78.2_{\pm1.5}$ | $48.4_{\pm2.9}$ | $82.5_{\pm0.9}$ | $63.8_{\pm0.9}$ | $73.3_{\pm0.7}$ | $51.9_{\pm1.1}$ | $89.0_{\pm1.0}$ | $44.1_{\pm1.4}$ | $80.4_{\pm1.4}$ | $22.3_{\pm1.5}$ |
| PTA [45] | $80.1_{\pm0.2}$ | $55.2_{\pm0.4}$ | $82.9_{\pm0.4}$ | $62.6_{\pm0.8}$ | $71.3_{\pm0.4}$ | $51.4_{\pm0.7}$ | $91.1_{\pm1.5}$ | $51.0_{\pm1.5}$ | $81.6_{\pm1.7}$ | $26.3_{\pm1.0}$ |
| GCN [3] | $79.7_{\pm0.3}$ | $55.8_{\pm0.6}$ | $81.4_{\pm0.8}$ | $59.3_{\pm1.1}$ | $70.9_{\pm0.8}$ | $49.8_{\pm0.6}$ | $91.0_{\pm1.2}$ | $52.0_{\pm1.0}$ | $82.4_{\pm1.5}$ | $27.0_{\pm1.5}$ |
| +LPA [12] | $79.6_{\pm0.6}$ | $53.5_{\pm0.9}$ | $81.7_{\pm0.7}$ | $60.3_{\pm1.5}$ | $71.0_{\pm0.6}$ | $50.2_{\pm1.0}$ | $91.3_{\pm1.2}$ | $52.9_{\pm2.0}$ | $83.7_{\pm1.4}$ | $28.5_{\pm2.4}$ |
| +GMNN [7] | $82.6_{\pm1.0}$ | $58.1_{\pm1.4}$ | $82.6_{\pm0.9}$ | $61.8_{\pm1.3}$ | $72.8_{\pm0.7}$ | $52.0_{\pm0.8}$ | $91.2_{\pm1.2}$ | $54.3_{\pm1.4}$ | $82.0_{\pm1.0}$ | $28.0_{\pm1.6}$ |
| +G³NN [8] | $80.9_{\pm0.7}$ | $56.9_{\pm1.1}$ | $82.5_{\pm0.4}$ | $62.3_{\pm0.6}$ | $73.9_{\pm0.7}$ | $53.1_{\pm1.0}$ | $90.7_{\pm1.1}$ | $53.0_{\pm2.0}$ | $82.1_{\pm1.2}$ | $28.1_{\pm2.1}$ |
| +CLGNN [13] | $81.7_{\pm0.5}$ | $57.8_{\pm0.7}$ | $81.9_{\pm0.5}$ | $61.8_{\pm0.8}$ | $72.0_{\pm0.7}$ | $51.6_{\pm0.9}$ | $91.1_{\pm1.0}$ | $53.4_{\pm1.8}$ | $83.3_{\pm1.2}$ | $28.5_{\pm1.4}$ |
| +DPM-SNC | $\mathbf{83.0}_{\pm0.9}$ | $\mathbf{59.2}_{\pm1.2}$ | $\mathbf{83.2}_{\pm0.5}$ | $\mathbf{63.1}_{\pm0.9}$ | $\mathbf{74.4}_{\pm0.5}$ | $\mathbf{53.6}_{\pm0.6}$ | $\mathbf{92.2}_{\pm0.8}$ | $\mathbf{55.3}_{\pm2.1}$ | $\mathbf{84.1}_{\pm1.3}$ | $\mathbf{29.7}_{\pm1.8}$ |
| GAT [21] | $79.1_{\pm0.5}$ | $55.8_{\pm0.5}$ | $81.5_{\pm0.6}$ | $61.3_{\pm0.9}$ | $71.0_{\pm0.8}$ | $50.8_{\pm1.0}$ | $90.8_{\pm1.0}$ | $50.8_{\pm1.9}$ | $83.1_{\pm1.6}$ | $27.8_{\pm2.2}$ |
| +LPA [12] | $78.7_{\pm1.1}$ | $56.0_{\pm1.2}$ | $81.5_{\pm0.9}$ | $60.7_{\pm0.8}$ | $71.3_{\pm0.9}$ | $50.1_{\pm0.9}$ | $91.3_{\pm0.8}$ | $52.7_{\pm2.1}$ | $\mathbf{84.4}_{\pm1.0}$ | $29.4_{\pm2.6}$ |
| +GMNN [7] | $79.6_{\pm0.8}$ | $57.0_{\pm0.7}$ | $82.3_{\pm0.7}$ | $62.2_{\pm0.8}$ | $71.7_{\pm0.9}$ | $51.4_{\pm0.9}$ | $91.4_{\pm1.0}$ | $53.1_{\pm1.6}$ | $83.3_{\pm2.0}$ | $29.1_{\pm1.8}$ |
| +G³NN [8] | $77.9_{\pm0.4}$ | $55.9_{\pm0.5}$ | $82.7_{\pm1.3}$ | $62.7_{\pm1.3}$ | $74.0_{\pm0.8}$ | $53.7_{\pm0.5}$ | $91.5_{\pm0.9}$ | $52.6_{\pm2.2}$ | $83.1_{\pm1.7}$ | $28.8_{\pm2.4}$ |
| +CLGNN [13] | $80.0_{\pm0.6}$ | $57.5_{\pm1.2}$ | $81.8_{\pm0.6}$ | $61.5_{\pm0.9}$ | $72.1_{\pm0.8}$ | $52.1_{\pm0.8}$ | $90.6_{\pm0.8}$ | $51.9_{\pm1.8}$ | $82.6_{\pm1.2}$ | $28.4_{\pm1.8}$ |
| +DPM-SNC | $\mathbf{81.7}_{\pm0.8}$ | $\mathbf{59.0}_{\pm1.1}$ | $\mathbf{83.8}_{\pm0.7}$ | $\mathbf{63.8}_{\pm0.7}$ | $\mathbf{74.3}_{\pm0.7}$ | $\mathbf{54.0}_{\pm0.9}$ | $\mathbf{92.0}_{\pm0.8}$ | $\mathbf{54.0}_{\pm2.4}$ | $84.2_{\pm1.2}$ | $\mathbf{30.0}_{\pm2.0}$ |
| GCNII [46] | $82.0_{\pm0.8}$ | $57.2_{\pm1.1}$ | $84.0_{\pm0.6}$ | $63.4_{\pm0.8}$ | $72.9_{\pm0.5}$ | $52.1_{\pm0.7}$ | $91.2_{\pm1.2}$ | $53.2_{\pm1.5}$ | $82.5_{\pm1.4}$ | $26.6_{\pm1.3}$ |
| +DPM-SNC | $\mathbf{83.8}_{\pm0.7}$ | $\mathbf{61.6}_{\pm0.9}$ | $\mathbf{85.3}_{\pm0.6}$ | $\mathbf{65.8}_{\pm0.7}$ | $\mathbf{74.1}_{\pm0.5}$ | $\mathbf{54.1}_{\pm0.9}$ | $\mathbf{92.8}_{\pm1.1}$ | $\mathbf{54.2}_{\pm1.2}$ | $\mathbf{84.4}_{\pm1.8}$ | $\mathbf{29.2}_{\pm1.1}$ |

The results, presented in Table 1, demonstrate that our method significantly improves accuracy compared to the baselines when labeled nodes are scattered. This highlights the superiority of DPM-SNC in considering label dependencies. Furthermore, our method also excels in the localized labeled nodes scenario, while the other baselines fail. These results can be attributed to the capabilities of DPMs for capturing long-range dependencies through iterative reverse diffusion steps.

**Homophilic graph.** In this experiment, we consider five datasets: Pubmed, Cora, and Citeseer [22]; Photo and Computer [23]. For all datasets, 20 nodes per class are used for training, and the remaining nodes are used for validation and testing. Detailed data statistics are in Appendix D.

We compare DPM-SNC with conventional GNN, and structured prediction baselines. We compare with label propagation-based methods: LP [42], C&S [43], BPN [44], and PTA [45]. We also consider G³NN, GMNN, and CLGNN. We describe the detailed experimental setup in Appendix E.2. As the backbone network, we consider GCN and GAT [21]. We further evaluate our algorithm with the recently developed GCNII [46]. For all the datasets, we evaluate the node-level accuracy. We also report the subgraph-level accuracy, which measures the ratio of nodes with all neighboring nodes being correctly classified. The performance is measured using ten different random seeds.

The results are presented in Table 2. Our method outperforms the structured prediction-specialized baselines in both node-label and subgraph-level accuracy. These results highlight the superiority of DPM-SNC in solving real-world node classification problems. Furthermore, even when we combine with GCNII [46], DPM-SNC consistently improves performance regardless of the backbone network.

Table 3: The transductive node classification accuracy on heterophilic graphs. **Bold** numbers indicate the best score.

| | Empire | Rating |
|---|---|---|
| H₂GCN [47] | $60.11_{\pm0.52}$ | $36.47_{\pm0.23}$ |
| CPGNN [48] | $63.96_{\pm0.62}$ | $39.79_{\pm0.77}$ |
| GPR-GNN [49] | $64.85_{\pm0.27}$ | $44.88_{\pm0.34}$ |
| FSGNN [50] | $79.92_{\pm0.56}$ | $52.74_{\pm0.83}$ |
| GloGNN [51] | $59.63_{\pm0.69}$ | $36.89_{\pm0.14}$ |
| FAGCN [52] | $65.22_{\pm0.56}$ | $44.12_{\pm0.30}$ |
| GBK-GNN [53] | $74.57_{\pm0.47}$ | $45.98_{\pm0.71}$ |
| JacobiConv [54] | $71.14_{\pm0.42}$ | $43.55_{\pm0.48}$ |
| GCN [3] | $73.69_{\pm0.74}$ | $48.70_{\pm0.63}$ |
| SAGE [4] | $85.74_{\pm0.67}$ | $53.63_{\pm0.39}$ |
| GAT [21] | $80.87_{\pm0.30}$ | $49.09_{\pm0.63}$ |
| GAT-sep [24] | $88.75_{\pm0.41}$ | $52.70_{\pm0.62}$ |
| GT [55] | $86.51_{\pm0.73}$ | $51.17_{\pm0.66}$ |
| GT-sep [24] | $87.32_{\pm0.39}$ | $52.18_{\pm0.80}$ |
| DPM-SNC | $\mathbf{89.52}_{\pm0.46}$ | $\mathbf{54.66}_{\pm0.39}$ |

**Heterophilic graph.** To validate whether our framework can also consider heterophily dependencies, we consider recently proposed heterophilic graph datasets: Empire and Ratings [24], where most heterophily-specific GNNs fail to solve. In Table 3, we compare our method with six GNNs:

Table 4: The inductive node classification performance. N-Acc., G-Acc., and F1 denote the node-level accuracy, graph-level accuracy, and micro-F1 score, respectively. **Bold** numbers indicate the best performance among the structured prediction methods using the same GNN.

| | Pubmed | | Cora | | Citeseer | | PPI |
|---|---|---|---|---|---|---|---|
| Method | N-Acc. | G-Acc. | N-Acc. | G-Acc. | N-Acc. | G-Acc. | F1 |
| GCN [3] | $80.25_{\pm 0.42}$ | $54.58_{\pm 0.51}$ | $83.36_{\pm 0.43}$ | $59.67_{\pm 0.51}$ | $76.37_{\pm 0.35}$ | $49.84_{\pm 0.47}$ | $99.15_{\pm 0.03}$ |
| +G$^3$NN [8] | $80.32_{\pm 0.30}$ | $53.93_{\pm 0.71}$ | $83.60_{\pm 0.25}$ | $59.78_{\pm 0.47}$ | $76.34_{\pm 0.37}$ | $50.76_{\pm 0.47}$ | $99.33_{\pm 0.02}$ |
| +CLGNN [13] | $80.22_{\pm 0.45}$ | $53.98_{\pm 0.54}$ | $83.45_{\pm 0.34}$ | $60.24_{\pm 0.38}$ | $75.71_{\pm 0.40}$ | $50.51_{\pm 0.38}$ | $99.22_{\pm 0.04}$ |
| +SPN [9] | $\mathbf{80.78}_{\pm 0.34}$ | $54.91_{\pm 0.40}$ | $83.85_{\pm 0.60}$ | $60.35_{\pm 0.57}$ | $76.25_{\pm 0.48}$ | $51.02_{\pm 1.06}$ | $99.35_{\pm 0.02}$ |
| +DPM-SNC | $80.58_{\pm 0.41}$ | $\mathbf{55.16}_{\pm 0.43}$ | $\mathbf{84.09}_{\pm 0.27}$ | $\mathbf{60.88}_{\pm 0.36}$ | $\mathbf{77.01}_{\pm 0.49}$ | $\mathbf{51.44}_{\pm 0.56}$ | $\mathbf{99.46}_{\pm 0.02}$ |
| GAT [21] | $80.10_{\pm 0.45}$ | $54.38_{\pm 0.54}$ | $79.71_{\pm 1.41}$ | $56.66_{\pm 1.40}$ | $74.91_{\pm 0.22}$ | $49.87_{\pm 0.44}$ | $99.54_{\pm 0.01}$ |
| +G$^3$NN [8] | $79.88_{\pm 0.62}$ | $54.66_{\pm 0.29}$ | $81.19_{\pm 0.45}$ | $58.68_{\pm 0.38}$ | $75.45_{\pm 0.26}$ | $50.86_{\pm 0.46}$ | $99.56_{\pm 0.01}$ |
| +CLGNN [13] | $80.23_{\pm 0.40}$ | $54.51_{\pm 0.36}$ | $81.38_{\pm 0.55}$ | $58.81_{\pm 0.61}$ | $75.45_{\pm 0.36}$ | $50.66_{\pm 0.45}$ | $99.55_{\pm 0.01}$ |
| +SPN [9] | $79.95_{\pm 0.34}$ | $\mathbf{54.82}_{\pm 0.33}$ | $81.61_{\pm 0.31}$ | $59.17_{\pm 0.31}$ | $75.41_{\pm 0.35}$ | $51.04_{\pm 0.53}$ | $99.46_{\pm 0.02}$ |
| +DPM-SNC | $\mathbf{80.26}_{\pm 0.37}$ | $54.26_{\pm 0.47}$ | $\mathbf{81.79}_{\pm 0.46}$ | $\mathbf{59.55}_{\pm 0.49}$ | $\mathbf{76.46}_{\pm 0.60}$ | $\mathbf{52.05}_{\pm 0.71}$ | $\mathbf{99.63}_{\pm 0.01}$ |

GCN, SAGE [4], GAT, GAT-sep [24], GT [55], and GT-sep [24]. We also compare with eight heterophily-specific GNNs: H$_2$GCN [47], CPGNN [48], GPR-GNN [49], FSGNN [50], GloGNN [51], FAGCN [52], GBK-GNN [53], and JacobiConv [54]. We employ GAT-sep as a backbone network of DPM-SNC. For all the baselines, we use the numbers reported by Platonov et al. [24].

Table 3 shows that our method again achieves competitive performance on heterophilic graphs. While existing heterophily-specific GNNs do not perform well on these datasets [24], our method shows improved performance stems from the extraordinary ability for considering label dependencies involving heterophily label dependencies.

## 6.2 Inductive setting

We further show that our DPM-SNC works well not only in transductive settings but also in inductive settings which involve inductive node classification and graph algorithmic reasoning. We provide the details of our implementation in Appendix C.2.

**Inductive node classification.** Following Qu et al. [9], we conduct experiments on both small-scale and large-scale graphs. We construct small-scale graphs from Pubmed, Cora, and Citeseer, and construct large-scale graphs from PPI [25]. The detailed data statistics are described in Appendix D.

We compare our DPM-SNC with the conventional GNN and three structured node classification methods: G3NN, CLGNN, and SPN [9]. As the backbone network of each method, we consider GCN and GAT. We evaluate node-level accuracy across all datasets and supplement it with additional metrics: graph-level accuracy for small-scale graphs and micro-F1 score for large-scale graphs. The graph-level accuracy measures the ratio of graphs with where all the predictions are correct. We report the performance measured using ten and five different random seeds for small-scale and large-scale graphs, respectively. The detailed experimental setup is described in Appendix E.3.

We report the results in Table 4. Here, DPM-SNC shows competitive results compared to all the baselines except for Pubmed. These results suggest that the DPM-SNC also solves inductive node classification effectively, thanks to their capability for learning joint dependencies.

**Algorithmic reasoning.** We also evaluate our DPM-SNC to predict the outcomes of graph algorithms, e.g., computing the shortest path between two nodes. Solving such tasks using GNNs has gained much attention since it builds connections between deep learning and classical computer science algorithms. Here, we show that the capability of DPM-SNC to make a structured prediction even brings benefits to solving the reasoning tasks by a deep understanding between algorithmic elements.

We evaluate the performance of our DPM-SNC on three graph algorithmic reasoning benchmarks proposed by Du et al. [26]: edge copy, connected component, and shortest path. The detailed data statistics are described in Appendix D. Here, we evaluate performance on graphs with ten nodes. Furthermore, we also use graphs with 15 nodes to evaluate generalization capabilities. We report the performance using element-wise mean square error.

Table 5: Performance on graph algorithmic reasoning tasks. **Bold** numbers indicate the best performance. Same-MSE and Large-MSE indicate the performance on ten, and 15 nodes, respectively.

| Method | Edge copy | | Connected components | | Shortest path | |
|---|---|---|---|---|---|---|
| | Same-MSE | Large-MSE | Same-MSE | Large-MSE | Same-MSE | Large-MSE |
| Feedforward | 0.3016 | 0.3124 | 0.1796 | 0.3460 | 0.1233 | 1.4089 |
| Recurrent [56] | 0.3015 | 0.3113 | 0.1794 | 0.2766 | 0.1259 | 0.1083 |
| Programmatic [57] | 0.3053 | 0.4409 | 0.2338 | 3.1381 | 0.1375 | 0.1290 |
| Iterative feedforward [58] | 0.6163 | 0.6498 | 0.4908 | 1.2064 | 0.4588 | 0.7688 |
| IREM [26] | 0.0019 | **0.0019** | 0.1424 | 0.2171 | 0.0274 | 0.0464 |
| DPM-SNC | **0.0011** | 0.0038 | **0.0724** | **0.1884** | **0.0138** | **0.0286** |

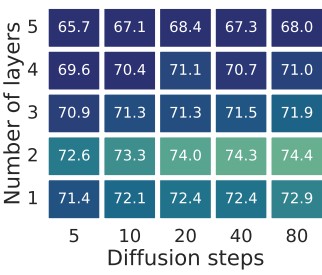 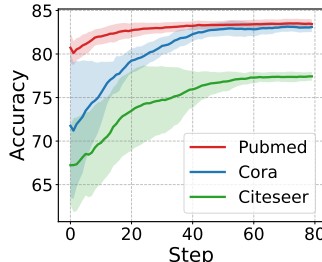 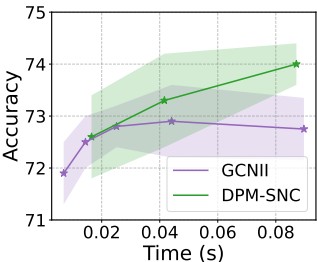

Figure 5: Accuracy with varying GNN layers and diffusion steps.

Figure 6: Accuracy for changes in diffusion steps.

Figure 7: Accuracy with varying inference time.

We compare our method with five methods reported by Du et al. [26], including a feedforward neural network, recurrent neural network [56], Pondernet [57], iterative feedforward [58], and IREM [26]. For all the baselines, we use the numbers reported by Du et al. [26]. As these tasks are defined on edge-wise targets, we modify DPM-SNC to make edge-wise predictions. We describe the detailed experimental setup in Appendix E.4.

We report the results in Table 5. Our DPM-SNC outperforms the considered baselines for five out of the six tasks. These results suggest that the diffusion model can easily solve algorithmic reasoning tasks thanks to its superior ability to make structured predictions.

## 6.3 Ablation studies

Here, we conduct three ablation studies to empirically analyze our framework. All the results are measured over ten different random seeds.

**Diffusion steps vs. number of GNN layers.** We first verify that the performance gains in DPM-SNC mainly stem from the reverse diffusion process which learns the joint dependency between labels. To this end, we vary the number of diffusion steps along with the number of GNN layers. We report the corresponding results in Figure 5. One can observe that increasing the number of diffusion steps provides a non-trivial improvement in performance, which cannot be achieved by just increasing the number of GNN layers.

**Accuracy over diffusion steps.** We investigate whether the iteration in the reverse process progressively improves the quality of predictions. In Figure 6, we plot the changes in node-level accuracy in the reverse process as the number of iterations increases. The results confirm that the iterative inference process gradually increases accuracy, eventually reaching convergence.

**Running time vs. performance.** Here, we investigate whether the DPM-SNC can make a good trade-off between running time and performance. In Figure 7, we compare the change in accuracy of DPM-SNC with GCNII over the inference time on Citeseer by changing the number of layers and diffusion steps for DPM-SNC and GCNII, respectively. The backbone network of DPM-SNC is a two-layer GCN. One can observe that our DPM-SNC shows competitive performances compared to the GCNII at a similar time. Also, while increasing the inference times of the GCNII does not enhance performance, DPM-SNC shows further performance improvement.

# 7 Conclusion and Discussion

In this paper, we propose diffusion probabilistic models for solving structured node classification (DPM-SNC). Extensive experiments on both transductive and inductive settings show that DPM-SNC outperforms existing structured node classification methods. An interesting avenue for future work is the study of characterizing the expressive power of GNNs to make structured predictions, i.e., the ability to learn complex dependencies between node-wise labels.

**Limitations.** Our DPM-SCN makes a trade-off between accuracy and inference time through the number of diffusion steps. Therefore, we believe accelerating our framework with faster diffusion-based models, e.g., denoising diffusion implicit models (DDIM) [59] is an interesting future work.

## Acknowledgements

This work partly was supported by Institute of Information & communications Technology Planning & Evaluation (IITP) grant funded by the Korea government(MSIT) (No. IITP-2019-0-01906, Artificial Intelligence Graduate School Program(POSTECH)), the National Research Foundation of Korea(NRF) grant funded by the Korea government(MSIT) (No. 2022R1C1C1013366), and Basic Science Research Program through the National Research Foundation of Korea(NRF) funded by the Ministry of Education(2022R1A6A1A03052954).

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

# A Details of DPM on partially labeled graphs

## A.1 Derivation of variational lower bound

In this section, we provide a detailed derivation of the variational lower bound in Equation (1).

$$
\begin{aligned}
\log p_{\boldsymbol{\theta}}(\boldsymbol{y}_L|G) &= \log \mathbb{E}_{p_{\boldsymbol{\theta}}(\boldsymbol{y}_U, \boldsymbol{y}^{(1:T)}|G)} \left[ p_{\boldsymbol{\theta}}(\boldsymbol{y}_L|G, \boldsymbol{y}_U, \boldsymbol{y}^{(1:T)}) \right] \\
&= \log \mathbb{E}_{q(\boldsymbol{y}_U, \boldsymbol{y}^{(1:T)}|\boldsymbol{y}_L)} \left[ \frac{p_{\boldsymbol{\theta}}(\boldsymbol{y}, \boldsymbol{y}^{(1:T)}|G)}{q(\boldsymbol{y}_U, \boldsymbol{y}^{(1:T)}|\boldsymbol{y}_L)} \right] \\
&\geq \mathbb{E}_{q(\boldsymbol{y}_U, \boldsymbol{y}^{(1:T)}|\boldsymbol{y}_L)} \left[ \log p_{\boldsymbol{\theta}}(\boldsymbol{y}, \boldsymbol{y}^{(1:T)}|G) - q(\boldsymbol{y}_U, \boldsymbol{y}^{(1:T)}|\boldsymbol{y}_L) \right] \\
&= \mathbb{E}_{q(\boldsymbol{y}_U, \boldsymbol{y}^{(1:T)}|\boldsymbol{y}_L)} \left[ \log p_{\boldsymbol{\theta}}(\boldsymbol{y}, \boldsymbol{y}^{(1:T)}|G) - q(\boldsymbol{y}^{(1:T)}|\boldsymbol{y}) - \log q(\boldsymbol{y}_U|\boldsymbol{y}_L) \right] \\
&= \mathbb{E}_{q(\boldsymbol{y}_U|\boldsymbol{y}_L)} \left[ \mathbb{E}_{q(\boldsymbol{y}^{(1:T)}|\boldsymbol{y})} [\log p_{\boldsymbol{\theta}}(\boldsymbol{y}, \boldsymbol{y}^{(1:T)}|G) - q(\boldsymbol{y}^{(1:T)}|\boldsymbol{y})] - \log q(\boldsymbol{y}_U|\boldsymbol{y}_L) \right].
\end{aligned}
$$

## A.2 Parameterization

Here, we provide more detailed parameterization for DPM-SNC. Following the Gaussian diffusion, we parameterize $p(\boldsymbol{y}^T)$ and $p_{\boldsymbol{\theta}}(\boldsymbol{y}^{(t-1)}|G, \boldsymbol{y}^{(t)})$ as $\mathcal{N}(\boldsymbol{y}^{(T)}; \mathbf{0}, \mathbf{I})$ and $\mathcal{N}(\boldsymbol{y}^{(t-1)}; \boldsymbol{\mu}_{\boldsymbol{\theta}}(\boldsymbol{y}^{(t)}, G, t), \sigma_t^2)$, respectively. Here, we set $\sigma_t^2$ to $\beta_t$. We also define $\mu_{\boldsymbol{\theta}}(\boldsymbol{y}^{(t)}, G, t)$ as follows:

$$
\mu_{\boldsymbol{\theta}}(\boldsymbol{y}^{(t)}, G, t) = \frac{1}{\sqrt{\alpha_t}} \left( \boldsymbol{y}^{(t)} - \frac{\beta_t}{\sqrt{1-\bar{\alpha}_t}} \epsilon_{\boldsymbol{\theta}}(\boldsymbol{y}^{(t)}, G, t) \right), \tag{2}
$$

where $\alpha_t = 1 - \beta_t$ and $\bar{\alpha}_t = \prod_{i=1}^{t} \alpha_i$. We implement the residual function $\epsilon_{\boldsymbol{\theta}}(\boldsymbol{y}^{(t)}, G, t)$ through a GNN. The implementation details are described in Appendix C.

## A.3 Detailed training objective

We describe the detailed training objective of $\mathcal{L}_{\text{VLB}}$ for optimization. We first rewrite $\mathcal{L}_{\text{VLB}}$ as follows:

$$
\begin{aligned}
&\mathcal{L}_{\text{VLB}} \\
&= \mathbb{E}_{q(\boldsymbol{y}_U|\boldsymbol{y}_L)} \left[ \mathbb{E}_{q(\boldsymbol{y}^{(1:T)}|\boldsymbol{y})} \left[ \log p_{\boldsymbol{\theta}}(\boldsymbol{y}, \boldsymbol{y}^{(1:T)}|G) - \log q(\boldsymbol{y}^{(1:T)}|\boldsymbol{y}) \right] - \log q(\boldsymbol{y}_U|\boldsymbol{y}_L) \right] \\
&= \mathbb{E}_{q(\boldsymbol{y}_U|\boldsymbol{y}_L)} \left[ \mathbb{E}_{q(\boldsymbol{y}^{(1:T)}|\boldsymbol{y})} \left[ \sum_{t=1}^{T} \log \frac{p_{\boldsymbol{\theta}}(\boldsymbol{y}^{(t-1)}|G, \boldsymbol{y}^{(t)})}{q(\boldsymbol{y}^{(t)}|\boldsymbol{y}^{(t-1)})} + \log p(\boldsymbol{y}^{(T)}) \right] - \log q(\boldsymbol{y}_U|\boldsymbol{y}_L) \right] \\
&= \mathbb{E}_{q(\boldsymbol{y}_U|\boldsymbol{y}_L)} \left[ \mathbb{E}_{q(\boldsymbol{y}^{(1:T)}|\boldsymbol{y})} \left[ \sum_{t=1}^{T} \log \frac{p_{\boldsymbol{\theta}}(\boldsymbol{y}^{(t-1)}|G, \boldsymbol{y}^{(t)})}{q(\boldsymbol{y}^{(t-1)}|\boldsymbol{y}^{(t)}, \boldsymbol{y})} \frac{q(\boldsymbol{y}^{(t-1)}|\boldsymbol{y})}{q(\boldsymbol{y}^{(t)}|\boldsymbol{y})} + \log p(\boldsymbol{y}^{(T)}) \right] - \log q(\boldsymbol{y}_U|\boldsymbol{y}_L) \right] \\
&= \mathbb{E}_{q(\boldsymbol{y}_U|\boldsymbol{y}_L)} \left[ \mathbb{E}_{q(\boldsymbol{y}^{(1:T)}|\boldsymbol{y})} \left[ \sum_{t=1}^{T} \log \frac{p_{\boldsymbol{\theta}}(\boldsymbol{y}^{(t-1)}|G, \boldsymbol{y}^{(t)})}{q(\boldsymbol{y}^{(t-1)}|\boldsymbol{y}^{(t)}, \boldsymbol{y})} + \log \frac{p(\boldsymbol{y}^{(T)})}{q(\boldsymbol{y}^{(T)}|\boldsymbol{y})} \right] - \log q(\boldsymbol{y}_U|\boldsymbol{y}_L) \right] \\
&= \mathbb{E}_{q(\boldsymbol{y}_U|\boldsymbol{y}_L)} \left[ \sum_{t=1}^{T} \mathbb{E}_{q(\boldsymbol{y}^{(1:T)}|\boldsymbol{y})} \left[ \log \frac{p_{\boldsymbol{\theta}}(\boldsymbol{y}^{(t-1)}|G, \boldsymbol{y}^{(t)})}{q(\boldsymbol{y}^{(t-1)}|\boldsymbol{y}^{(t)}, \boldsymbol{y})} \right] + \mathbb{E}_{q(\boldsymbol{y}^{(T)}|\boldsymbol{y})} \left[ \log \frac{p(\boldsymbol{y}^{(T)})}{q(\boldsymbol{y}^{(T)}|\boldsymbol{y})} \right] - \log q(\boldsymbol{y}_U|\boldsymbol{y}_L) \right] \\
&= \mathbb{E}_{q(\boldsymbol{y}_U|\boldsymbol{y}_L)} \left[ \sum_{t=1}^{T} \mathcal{L}_{\text{DPM}}^{(t)} + C - \log q(\boldsymbol{y}_U|\boldsymbol{y}_L) \right],
\end{aligned}
$$

where $\mathcal{L}_{\text{DPM}}^{(t)}$ is a training objective for a $t$ step, and $C$ is a constant with respect to the parameters $\boldsymbol{\theta}$. Following Ho et al. [16], we simplify $\mathcal{L}_{\text{DPM}}^{(t)}$ with a residual function $\epsilon_{\boldsymbol{\theta}}(\boldsymbol{y}^{(t)}, G, t)$ in Equation (2).

$$
\mathcal{L}_{\text{DPM}}^{(t)} = C - \mathbb{E}_{\epsilon \sim \mathcal{N}(\mathbf{0}, \mathbf{I})} \left[ \frac{\beta_t^2}{2\sigma_t^2 \alpha_t (1-\bar{\alpha}_t)} \left\| \epsilon - \epsilon_{\boldsymbol{\theta}}(\sqrt{\bar{\alpha}_t}\boldsymbol{y} + \sqrt{1-\bar{\alpha}_t}\epsilon, G, t) \right\|_2^2 \right],
$$

where $C$ is a constant with respect to the parameters $\boldsymbol{\theta}$. An additional suggestion from Ho et al. [16] is to set all weights of the mean squared error to one instead of $\beta_t^2/2\sigma_t^2\alpha_t(1-\bar{\alpha}_t)$, and we follow this suggestion in this paper.

## A.4 Manifold-constrained sampling

---

**Algorithm 2** Manifold-constrained sampling

---

1: **Input:** Graph $G$, node attributes $\boldsymbol{x}$, labels $\boldsymbol{y}_L$, and temperature of randomness $\tau^2$.
2: Get $\boldsymbol{y}^{(T)} \sim \mathcal{N}(\boldsymbol{y}^{(T)}; \boldsymbol{0}, \tau^2 \mathbf{I})$        ▷ *Initial sampling*
3: **for** $t = T - 1, \ldots, 0$ **do**
4:   Get $\boldsymbol{z} \sim \mathcal{N}(\boldsymbol{z}; \boldsymbol{0}, \tau^2 \mathbf{I})$
5:   Set $\tilde{\boldsymbol{y}}^{(t)} \leftarrow \frac{1}{\sqrt{\alpha_t}} \left( \boldsymbol{y}^{(t+1)} - \frac{\beta_{t+1}}{\sqrt{1-\bar{\alpha}_{t+1}}} \boldsymbol{\epsilon_\theta}(\boldsymbol{y}^{(t+1)}, G, t+1) \right) + \sigma_{t+1} \boldsymbol{z}$
6:   Set $\hat{\boldsymbol{y}}^{(t+1)} \leftarrow \frac{1}{\sqrt{\bar{\alpha}_{t+1}}} \left( \boldsymbol{y}^{(t+1)} - \frac{1-\bar{\alpha}_{t+1}}{\sqrt{1-\bar{\alpha}_{t+1}}} \boldsymbol{\epsilon_\theta}(\boldsymbol{y}^{(t+1)}, G, t+1) \right)$
7:   Set $\bar{\boldsymbol{y}}^{(t)} \leftarrow \left( \tilde{\boldsymbol{y}}^{(t)} - \gamma \frac{\partial}{\partial \boldsymbol{y}^{(t+1)}} \left\| \boldsymbol{y}_L - \hat{\boldsymbol{y}}_L^{(t+1)} \right\|_2^2 \right)$    ▷ *Manifold-constrained gradient*
8:   Set $\boldsymbol{y}_U^{(t)} \leftarrow \bar{\boldsymbol{y}}_U^{(t)}$
9:   Get $\boldsymbol{z} \sim \mathcal{N}(\boldsymbol{z}; \boldsymbol{0}, \tau^2 \mathbf{I})$
10:   Set $\boldsymbol{y}_L^{(t)} \leftarrow \sqrt{\bar{\alpha}_t} \boldsymbol{y}_L + \sqrt{1 - \bar{\alpha}_t} \boldsymbol{z}$       ▷ *Projection step*
11:   Set $\boldsymbol{y}^{(t)} \leftarrow \boldsymbol{y}_L^{(t)} \cup \boldsymbol{y}_U^{(t)}$
12: **end for**
13: **return** $\boldsymbol{y}_U^{(0)}$

---

To sample $\boldsymbol{y}_U$ from $p_{\boldsymbol{\theta}}(\boldsymbol{y}_U | G, \boldsymbol{y}_L)$, we use a manifold-constrained sampling proposed by Chung et al. [18]. Here, the update rule of the reverse process for $t = 0, \ldots, T - 1$ is defined as follows:

$$\tilde{\boldsymbol{y}}^{(t)} = \frac{1}{\sqrt{\alpha_t}} \left( \boldsymbol{y}^{(t+1)} - \frac{\beta_{t+1}}{\sqrt{1 - \bar{\alpha}_{t+1}}} \boldsymbol{\epsilon_\theta}(\boldsymbol{y}^{(t+1)}, G, t+1) \right) + \sigma_{t+1} \boldsymbol{z}, \tag{3}$$

$$\boldsymbol{y}_U^{(t)} = \bar{\boldsymbol{y}}_U^{(t)}, \qquad \bar{\boldsymbol{y}}^{(t)} = \tilde{\boldsymbol{y}}^{(t)} - \gamma \frac{\partial}{\partial \boldsymbol{y}^{(t+1)}} \left\| \boldsymbol{y}_L - \hat{\boldsymbol{y}}_L^{(t+1)} \right\|_2^2, \tag{4}$$

$$\boldsymbol{y}_L^{(t)} = \sqrt{\bar{\alpha}_t} \boldsymbol{y}_L + \sqrt{1 - \bar{\alpha}_t} \boldsymbol{z}, \tag{5}$$

where $\boldsymbol{z}$ is sampled from $\mathcal{N}(\boldsymbol{z}; \boldsymbol{0}, \tau^2 \mathbf{I})$. The Equation (3) is a temporal reverse diffusion step before applying the manifold-constrained samplings. The Equation (4) applies the manifold-constrained gradient $\gamma \frac{\partial}{\partial \boldsymbol{y}^{(t+1)}} \| \boldsymbol{y}_L - \hat{\boldsymbol{y}}_L^{(t+1)} \|_2^2$. Here, $\hat{\boldsymbol{y}}^{(t+1)}$ is the label estimate in $t + 1$ steps defined as follows:

$$\hat{\boldsymbol{y}}^{(t+1)} = \frac{1}{\sqrt{\bar{\alpha}_{t+1}}} \left( \boldsymbol{y}^{(t+1)} - \frac{1 - \bar{\alpha}_{t+1}}{\sqrt{1 - \bar{\alpha}_{t+1}}} \boldsymbol{\epsilon_\theta}(\boldsymbol{y}^{(t+1)}, G, t+1) \right),$$

where $\gamma$ is a hyper-parameter. We set $\gamma$ to $1/\| \boldsymbol{y}_L - \hat{\boldsymbol{y}}_L^{(t+1)} \|_2^2$. The Equation (5) is a projection step. Additionally, we introduce a parameter $\tau$ to control the randomness; when we set $\tau$ to zero, the modified reverse step becomes deterministic. This allows us to control the randomness in obtaining samples. We describe the detailed sampling algorithm in Algorithm 2.

# B WL test and GNN's expressiveness

In this section, we provide the proof of Theorem 1 in detail.

## B.1 Preliminaries

---
**Algorithm 3** 1-dimensional Weisfeiler-Lehman algorithm

---
1: **Input:** Graph $G = (\mathcal{V}^G, \mathcal{E}^G, \boldsymbol{x}^G)$ and the number of iterations $T$.
2: **Output:** Color mapping $\chi_G : \mathcal{V}^G \to \mathcal{C}$.
3: **Initialize:** Let $\chi_G^0(v) \leftarrow \text{hash}(x_v)$ for $v \in \mathcal{V}^G$.
4: **for** $t = 1, \ldots, T$ **do**
5:     **for** each $v \in \mathcal{V}^G$ **do**
6:         Set $\chi_G^t(v) \leftarrow \text{hash}(\chi_G^{t-1}(v), \{\!\{\chi_G^{t-1}(u) : u \in \mathcal{N}_G(v)\}\!\})$
7:     **end for**
8: **end for**
9: **Return:** $\chi_G^T$

---

We denote set as $\{\}$, and multiset as $\{\!\{\}\!\}$, which is a set allowing duplicate elements. We represent the cardinality of set or multiset $\mathcal{S}$ as $|\mathcal{S}|$. A graph is denoted as $G = (\mathcal{V}^G, \mathcal{E}^G, \boldsymbol{x}^G)$, where $\mathcal{V}^G$ stands for the set of nodes, $\mathcal{E}^G$ for the set of edges, and $\boldsymbol{x}^G = \{x_v^G : v \in \mathcal{V}^G\}$ for the node-wise attributes. We also use $\mathcal{N}_G(v) := \{u \in \mathcal{V}^G : \{v, u\} \in \mathcal{E}^G\}$ to denote the set of neighbor nodes of node $v$ in graph $G$. We abbreviate a set of integers using the notation $[m] := \{0, \ldots, m\}$. We also assume the hash functions are all injective and denote them by $\text{hash}(\cdot)$.

**Definition 1** (Vertex coloring). *Vertex coloring $\chi_G(v)$ is an injective hash function that maps a vertex $v$ in graph $G$ to a color $c$ from an abstract color set $\mathcal{C}$. Next, with a slight abuse of notation, we let the graph color $\chi_G$ indicate the multiset of node colors in graph $G$, i.e., $\chi_G := \{\!\{\chi_G(v) : v \in \mathcal{V}^G\}\!\}$.*

Now we explain the 1-dimensional Weisfeiler-Lehman (1-WL) algorithm [19], a classical algorithm to distinguish non-isomorphic graphs. At a high level, the 1-WL test iteratively updates node colors based on their neighbors until a stable coloring is reached. To be specific, given a vertex $v$, the initial node color $\chi_G^0(v)$ is set using an injective hash function on the node attribute $x_v$. At each iteration, each node color is refined based on the aggregation of neighbor node colors, e.g., $\{\!\{\chi_G^{t-1}(u) : u \in \mathcal{N}_G(v)\}\!\}$. At the final $T$-th iteration, the algorithm returns the graph color $\chi_G^T$. We provide a detailed description in Algorithm 3.

The 1-WL test allows to compare a pair of graphs $G, H$ using the refined graph colors $\chi_G^T, \chi_H^T$. If the two graph colors $\chi_G^T$ and $\chi_H^T$ are not equivalent, two graphs $G, H$ are guaranteed to be non-isomorphic. Otherwise, the test is inconclusive, i.e., the two graphs $G, H$ are possibly isomorphic. We provide an example run of the 1-WL test in Figure 8(a), where the test is inconclusive.

Related to the 1-WL algorithm, we first prove a characteristic of it that will be later used in our proof.

**Lemma 1.** *Consider running 1-WL on two graphs $G = (\mathcal{V}^G, \mathcal{E}^G, \boldsymbol{x}^G)$ and $H = (\mathcal{V}^H, \mathcal{E}^H, \boldsymbol{x}^H)$. If the initial graph colors of two graphs are distinct, i.e., $\chi_G^0 \neq \chi_H^0$, the respective outputs of the 1-WL are also distinct, i.e., $\chi_G^T \neq \chi_H^T$.*

*Proof.* We first prove that $\chi_G^t \neq \chi_H^t$ is satisfied when $\chi_G^{t-1} \neq \chi_H^{t-1}$. The rest of the proof is straightforward by induction on $t$. To be specific, the 1-WL updates $\chi_G^t$ and $\chi_H^t$ as follows:

$$\chi_G^t = \{\!\{\text{hash}(\chi_G^{t-1}(v), \{\!\{\chi_G^{t-1}(u) : u \in \mathcal{N}_G(v)\}\!\}) : v \in \mathcal{V}^G\}\!\},$$
$$\chi_H^t = \{\!\{\text{hash}(\chi_H^{t-1}(v), \{\!\{\chi_H^{t-1}(u) : u \in \mathcal{N}_H(v)\}\!\}) : v \in \mathcal{V}^H\}\!\}.$$

Since $\text{hash}(\cdot)$ is an injective function, $\chi_G^t$ and $\chi_H^t$ are distinct for $\chi_G^{t-1} \neq \chi_H^{t-1}$. $\qquad\square$

## B.2 AGG-WL test

Next, we describe the newly proposed AGG-WL test, which is an analog of our DPM-SNC. Similar to the 1-WL test, our AGG-WL test assigns node colors and iteratively refines them based on the

neighbor node colors. However, AGG-WL generates augmented graphs at initialization, creating multiple "views" on the graph with diverse initial vertex coloring. Then it applies 1-WL on each of the augmented graphs to obtain the augmented graph colors. Finally, AGG-WL aggregates the augmented graphs colors.

The complete algorithm is described in Algorithm 4. Given a graph $G = (\mathcal{V}^G, \mathcal{E}^G, \boldsymbol{x}^G)$ and the set of possible node-wise augmentations $\mathcal{Z}^{\mathcal{G}}$, the algorithm generates augmented graph $G^m$. In particular, the augmented graph $G^m$ is defined as follows:

$$G^m = (\mathcal{V}^G, \mathcal{E}^G, \tilde{\boldsymbol{x}}^{G^m}), \quad \tilde{\boldsymbol{x}}^{G^m} = \{(x_v^G \,\|\, z_v^m) : v \in \mathcal{V}\}, \quad \boldsymbol{z}^m = \{z_v^m : v \in \mathcal{V}\}, \quad \boldsymbol{z}^m \in \mathcal{Z}^G$$

where $\boldsymbol{z}^m$ is an augmentation method in $\mathcal{Z}^G$. The algorithm creates an augmented graph $G^m$ by node-wise concatenation of $\boldsymbol{z^m}$ to its node attributes $\boldsymbol{x}^G$, where $\tilde{\boldsymbol{x}}^{G^m}$ denotes the node-wise attribute augmented by $\boldsymbol{z}^m$. The symbol $\cdot\|\cdot$ denotes the concatenation of two elements. Furthermore, $\boldsymbol{z}^0$ is a unique token, where $G^0$ has the same information as $G$. Also, with a slight abuse of notation, we let the graph color returned by the AGG-WL $\chi_G^{\text{AGG}}$ indicate the multiset of augmented graph colors, i.e., $\chi_G^{\text{AGG}} := \{\!\!\{\chi_{G^m}^T : m \in [|\mathcal{Z}^G|]\}\!\!\}$. Here, $|\mathcal{Z}^G|$ denotes the cardinality of set $\mathcal{Z}^G$.

---

**Algorithm 4** Aggregation Weisfeiler-Lehman algorithm

1: **Input:** Graph $G = (\mathcal{V}^G, \mathcal{E}^G, \boldsymbol{x}^G)$, the number of iterations $T$, and the augmentation set $\mathcal{Z}^G$.
2: **Output:** Color mapping $\chi_G : \mathcal{V}^G \to \mathcal{C}$.
3: **Initialize:** Generate $|\mathcal{Z}^G|$ augmented graphs $G^m = (\mathcal{V}^G, \mathcal{E}^G, \boldsymbol{x}^{G^m})$ where $\tilde{\boldsymbol{x}}^{G^m} = \{(x_v^G \,\|\, z_v^m) : v \in \mathcal{V}^G\}$ for $\boldsymbol{z}^m \in \mathcal{Z}^G$. Let $\chi_{G^m}^0(v) \leftarrow \text{hash}(\tilde{\boldsymbol{x}}_v^{G^m})$ for $v \in \mathcal{V}^G, m \in [|\mathcal{Z}^G|]$.
4: **for** $t = 1, \ldots, T$ **do**
5:    **for** each $v \in V$ **do**
6:       **for** $m \in [|\mathcal{Z}^G|]$ **do**
7:          $\chi_{G^m}^t(v) \leftarrow \text{hash}(\chi_{G^m}^{t-1}(v), \{\!\!\{\chi_{G^m}^{t-1}(u) \; : u \in \mathcal{N}_{G^m}(v)\}\!\!\})$
8:       **end for**
9:    **end for**
10: **end for**
11: $\chi_G^{\text{AGG}} \leftarrow \{\!\!\{\chi_{G^m}^T : m \in [|\mathcal{Z}^G|]\}\!\!\}$
12: **Return:** $\chi_G^{\text{AGG}}$

---

Our contribution is establishing the connection between the AGG-WL and the DPM-SNC: aggregation of refined graph colors for augmented graphs and marginalization over latent variables. More detail between DPM-SNC and the AGG-WL test is described in Appendix B.3.2.

We note that our algorithm bears some similarities with several previous studies. The DS-WL and DSS-WL test [41] defined a modified WL-test based on the set of subgraphs (instead of augmented graphs) with a modified edge set $\mathcal{E}'$. Hang et al. [13] uses a collective algorithm to consider pseudo-labels as additional inputs to boost GNN expressiveness. However, they rely on the assumption that one can find an "optimal" pseudo-label to discriminate a pair of graphs, which may be hard to realize in practice. Zhange et al. [60] proposed labeling tricks that learns to capture dependence between nodes also by adding additional features, but mainly focus on link prediction in inductive settings.

## B.3 Proof of Theorem 1

Let us start by restating Theorem 1.

**Theorem 1.** *Let 1-WL-GNN be a GNN as powerful as the 1-WL test. Then, DPM-SNC using a 1-WL-GNN is strictly more powerful than the 1-WL-GNN in distinguishing non-isomorphic graphs.*

*Proof.* We divide the proof into two parts, Appendices B.3.1 and B.3.2. In Appendix B.3.1, we show that the AGG-WL test is strictly more powerful than the 1-WL test. In Appendix B.3.2, we show that DPM-SNC using a 1-WL-GNN is as powerful as the AGG-WL test. From these two proofs, one can conclude that DPM-SNC using a 1-WL-GNN is strictly more powerful than 1-WL-GNNs in distinguishing non-isomorphic graphs. □

### B.3.1 Expressiveness of AGG-WL test

In this subsection, we prove that the AGG-WL test is strictly more powerful that the 1-WL test. The high-level idea is showing that (i) the AGG-WL test is at least as powerful as the 1-WL test, i.e., any two graphs distinguishable by the 1-WL test is also distinguishable by the AGG-WL test, and (ii) there exist two non-isomorphic graphs that are indistinguishable by the 1-WL test, but distinguishable by our AGG-WL test.

**Lemma 2.** *AGG-WL is at least as powerful as WL in distinguishing non-isomorphic graphs, i.e., any two non-isomorphic graphs $G, H$ distinguishable by WL are also distinguishable by AGG-WL.*

*Proof.* We first note that both 1-WL and AGG-WL can discriminate two non-isomorphic graphs with distinct sizes in a straightforward way. Therefore, we focus on non-trivial cases for two graphs $G = (\mathcal{V}^G, \mathcal{E}^G, \boldsymbol{x}^G)$, $H = (\mathcal{V}^H, \mathcal{E}^H, \boldsymbol{x}^H)$ where the number of nodes and edges are same, i.e., $|\mathcal{V}^G| = |\mathcal{V}^H|, |\mathcal{E}^G| = |\mathcal{E}^H|$. The cardinality of the augmentation sets are also the same, $|\mathcal{Z}^G| = |\mathcal{Z}^H|$. We prove if these two graphs $G, H$ are distinguishable by the 1-WL, two graphs $G, H$ are also distinguishable by the AGG-WL, i.e., if $\chi_G^T \neq \chi_H^T$, then $\chi_G^{\text{AGG}} \neq \chi_H^{\text{AGG}}$.

First, since $\chi_G^T \neq \chi_H^T$ is satisfied, $\chi_{G^0}^T \neq \chi_{H^0}^T$ is trivial. Additionally, the initial graphs colors of graph augmented by the unique token and any other augmentations are distinct, i.e., $\chi_{G^0}^0 \neq \chi_{H^1}^0, \ldots, \chi_{H^{|\mathcal{Z}^H|}}^0$. This implies $\chi_{G^0}^T \notin \chi_H^{\text{AGG}} = \{\chi_{H^0}^T, \ldots, \chi_{H^{|\mathcal{Z}^H|}}^T\}$ according to Lemma 1 which shows distinct initial graph colors produce the distinct outputs of 1-WL. Since $\chi_{G^0}^T \in \chi_G^{\text{AGG}}$, one can see that $\chi_G^{\text{AGG}} \neq \chi_H^{\text{AGG}}$. $\qquad\square$

Next, we show that there exist two graphs $G, H$ indistinguishable by the 1-WL test, but distinguishable by the AGG-WL test. We first show a simple example from Figure 8(a), then for a family of circular skip link (CSL) graphs [61].

**Lemma 3.** *Two graphs $A, B$ are indistinguishable by the 1-WL, but distinguishable by the AGG-WL.*

*Proof.* In Figure 8(a), two graphs $A, B$ with 6 nodes are given. Here, the graph color computed by the 1-WL are equivalent, i.e., $\chi_G^T = \chi_H^T$ (two light blue nodes and four gray nodes). Therefore, two graphs $A, B$ are indistinguishable by the 1-WL.

Next, we prove that two graphs $A, B$ are distinguishable by the AGG-WL. Since Lemma 1 shows 1-WL always computes a distinct graph color from distinct initial graph colors, we only need to show that there exists augmented graphs $A^m, B^m$ with the same initial color has distinct graph colors computed by the 1-WL. Then, for two augmented graph sets $\{A^m : m \in [|\mathcal{Z}^{\mathcal{V}^A}|]\}$ and $\{B^m : m \in [|\mathcal{Z}^{\mathcal{V}^B}|]\}$, we can show that the graph colors computed by AGG-WL are always distinct.

We show a simple case, where augmented graphs with non-zero augmentation on one node results distinct graph colors computed by the AGG-WL. We denote "single binary node feature augmentation", $\boldsymbol{z}^i$ as the case where node $i$ is non-zero augmented and other nodes are augmented with zero, e.g., $\boldsymbol{z}^3 = \{0, 0, 1, 0, 0, 0\}$. In Figure 8(b) the non-zero augmented node $i$ is colored in blue, and others are colored transparent. After applying the 1-WL algorithm on each graph, we obtain the graph color $\chi_{A^i}^T$ and $\chi_{B^i}^T$ for $i \in \{1, \ldots, 6\}$. It is clear that $\{\!\{\chi_{A^i}^T : i \in \{1, \ldots, 6\}\}\!\} \neq \{\!\{\chi_{B^i}^T : i \in \{1, \ldots, 6\}\}\!\}$, thereby two graphs $A, B$ are distinguishable by AGG-WL. $\qquad\square$

**Circular skip link graphs.** CSL graph is denoted as $\text{CSL}(n, r)$, where $n$ is the number of nodes, .i.e., $V = \{0, \ldots, n-1\}$ and $r$ is the skip connection length. For $n$ and $r$, $r < n - 1$ must hold. There exists $2n$ edges, between node $i$ and $(i+1) \mod n$ forming a cycle, and between nodes $i$ and $(i+r) \mod n$ forming a skip link for $i \in \{0, \ldots, n-1\}$.

**Lemma 4.** *For $n \geq 8, r \in [3, n/2 - 1]$, two graphs $CSL(n, 2)$ and $CSL(n, r)$ are indistinguishable by the 1-WL, but distinguishable by the AGG-WL.*

*Proof.* We first give a brief proof of why $\text{CSL}(n, 2)$ and $\text{CSL}(n, r)$ are indistinguishable by the 1-WL, then prove they are distinguishable by AGG-WL. We consider a non-trivial case where the node attributes are all same, i.e., $\chi_G^0(v) = \chi_G^0(u)$ for $v, u \in \mathcal{V}^G$.

Let the initial color be $c_0$ for all the nodes for graphs $\text{CSL}(n, 2)$ and $\text{CSL}(n, r)$, $\chi_G^0(v) = c_0 \,\forall v \in \mathcal{V}^G, G \in \{\text{CSL}(n, 2), \text{CSL}(n, r)\}$. Then, for all nodes $v \in \mathcal{V}^G$, the color refinement process can

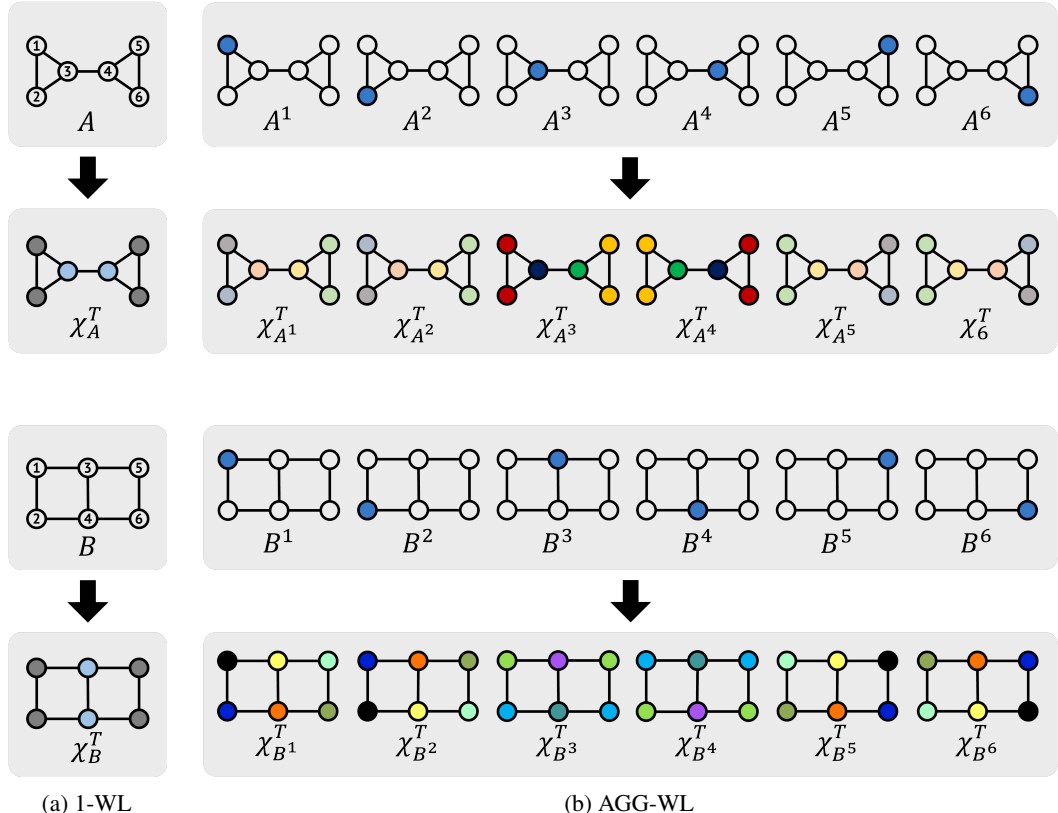

Figure 8: An example of two non-isomorphic graphs $A, B$ indistinguishable by 1-WL, but distinguishable by the AGG-WL. (a) Two graphs $A, B$ are associated with nodes sharing the same attribute. The 1-WL computes the same graph color for $A, B$, i.e., $\chi_A^T = \chi_B^T$. (b) Applying AGG-WL with "single binary node feature augmentation" to the graphs $A$ and $B$. This results in six cases for each graph. One can see that the multiset of augmented graph colors are different, i.e., $\{\!\{\chi_{A^i}^T : i \in \{1, \ldots, 6\}\}\!\} \neq \{\!\{\chi_{B^i}^T : i \in \{1, \ldots, 6\}\}\!\}$, thereby two graphs $A, B$ are distinguishable by the AGG-WL.

than be written as $\chi_G^1(v) = \text{hash}(c_0, \{\!\{c_0, c_0, c_0, c_0\}\!\})$. All nodes have identical colors $c_1$ for both graphs $\text{CSL}(n, 2), \text{CSL}(n, r)$, refining the initial graph color. Therefore $\text{CSL}(n, 2)$ and $\text{CSL}(n, r)$ are indistinguishable by the 1-WL.

Now we consider the AGG-WL. Again, we only need to show that there exists augmented graphs $A^i, B^j$ with the same initial color but having distinct graph colors refined by the 1-WL. Given symmetry, there is only one case of an augmented graph that adds a non-zero augmentation to one node. We use $v_i$ to denote the $i$-th node in graph and denote the augmented node as $v_0$. We denote each augmented graph as $\text{CSL}(n, 2)^1, \text{CSL}(n, r)^1$, and let the initial color $\chi_G^0(v_0) = c_1$, and $\forall i \in \{1, \ldots, n-1\}, \forall G \in \{\text{CSL}(n, 2)^1, \text{CSL}(n, r)^1\} \ \chi_G^0(v_i) = c_0$.

*Iteration 1.* We focus on the four nodes connected to $v_0$. Since $r \in [3, n/2 - 1]$, two node $v_{n-r}, v_r$ are distinct. The color refinement can be written as following:

- For $v \in \{v_1, v_2, v_{n-1}, v_{n-2}\}$, $\chi_{\text{CSL}(n,2)^1}^1(v) = \text{hash}(c_0, \{\!\{c_1, c_0, c_0, c_0\}\!\}) = c_2$.

- For $v \notin \{v_1, v_2, v_{n-1}, v_{n-2}\}$, $\chi_{\text{CSL}(n,2)^1}^1(v) = \text{hash}(c_0, \{\!\{c_0, c_0, c_0, c_0\}\!\}) = c_3$.

- For $v \in \{v_1, v_r, v_{n-1}, v_{n-r}\}$, $\chi_{\text{CSL}(n,r)^1}^1(v) = \text{hash}(c_0, \{\!\{c_1, c_0, c_0, c_0\}\!\}) = c_2$.

- For $v \notin \{v_1, v_r, v_{n-1}, v_{n-r}\}$, $\chi_{\text{CSL}(n,r)^1}^1(v) = \text{hash}(c_0, \{\!\{c_0, c_0, c_0, c_0\}\!\}) = c_3$.

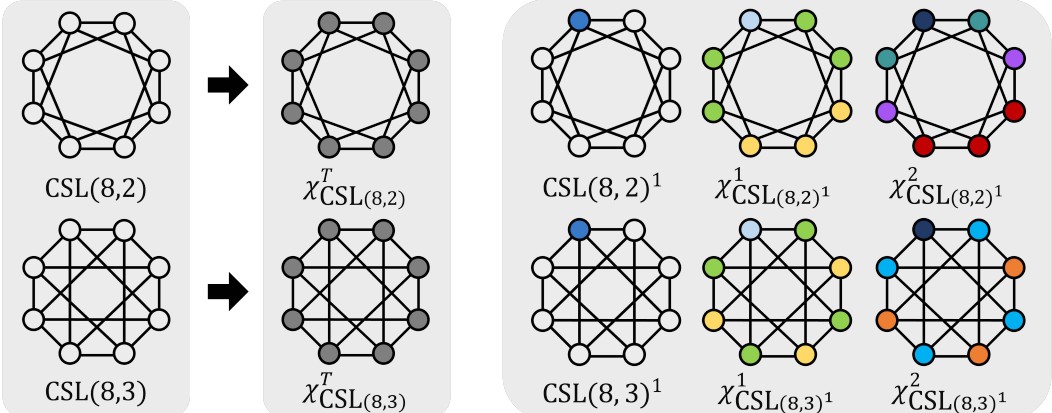

(a) Original graph and 1-WL coloring

(b) AGG-WL coloring on $CSL(8,2)^1, CSL(8,3)^1$

Figure 9: Example on two non-attribute $CSL(8,2), CSL(8,3)$ graph. (a) 1-WL returns same graph color for two graphs, i.e., $\chi_{CSL(8,2)}^T = \chi_{CSL(8,3)}^T$. (b) Considering symmetry, there is only one type of augmented graph that with non-zero augmentation on one node, we denoted the augmented graph as $CSL(8,2)^1$ and $CSL(8,3)^1$. In the process of 1-WL assigning colors to augmented graph, they result different graph color from the second iteration, and one can conclude $\chi_{CSL(8,2)^1}^T \neq \chi_{CSL(8,3)^1}^T$. Therefore $CSL(8,2), CSL(8,3)$ are distinguishable by the AGG-WL.

Since 1-WL showed $\chi_{CSL(n,2)}^1 = \chi_{CSL(n,r)}^1$ and augmented graphs showed $\chi_{CSL(n,2)^1}^1 = \chi_{CSL(n,r)^1}^1$, two graphs are indistinguishable.

*Iteration 2.* Again, we focus on four nodes connected to $v_0$. Here, we describe color refinement for nodes $v_1, v_{n-1}, v_2, v_{n-2}$, since the following is enough to prove two graph colors are distinct.

- For $v \in \{v_1, v_{n-1}\}$, $\chi_{CSL(n,2)^1}^2(v) = \text{hash}(c_2, \{\!\{c_1, c_2, c_2, c_3\}\!\}) = c_4$.

- For $v \in \{v_2, v_{n-2}\}$, $\chi_{CSL(n,2)^1}^2(u) = \text{hash}(c_2, \{\!\{c_1, c_2, c_3, c_3\}\!\}) = c_5$.

- For $v \in \{v_1, v_{n-1}\}$, $\chi_{CSL(n,r)^1}^2(v) = \text{hash}(c_2, \{\!\{c_1, c_3, c_3, c_3\}\!\}) = c_6$.

Since $c_6 \notin \chi_{CSL(n,2)^1}^2$, two graphs are distinguishable, i.e., $\chi_{CSL(n,2)^1}^t \neq \chi_{CSL(n,r)^1}^t$.

Then $\chi_{CSL(n,2)^1}^T \neq \chi_{CSL(n,r)^1}^T$ is satisfied as proved in Lemma 1, thereby two graphs $CSL(n,2)$ and $CSL(n,r)$ are distinguishable by the AGG-WL. $\qquad\square$

In Figure 9, we provide an example with $CSL(8,2)$ and $CSL(8,3)$, indistinguishable by the 1-WL but distinguishable the AGG-WL.

### B.3.2 Correspondence between DPM-SNC and AGG-WL

Now, we explain the connection between the DPM-SNC and the AGG-WL test. At a high-level, we show how DPM-SNC simulates the color refinement process of the AGG-WL. Our main idea stems from the marginalization in DPM-SNC, which can define the probability of the graph color $\chi$ with the marginalization over augmented graphs, i.e., $p_\theta(\chi|G) = \int p_\theta(\chi|G, z)p(z|G)dz$.[5] Specifically, DPM-SNC considers sample space of graph colors, where each member $\chi$ is obtained from $p_\theta(\chi|G, z)$ with an augmented graph $G^m$ represented by $(G, z^m)$. By construction, DPM-SNC considers graph colors refinements for multiple augmented graphs, similar to how AGG-WL works.

**Lemma 5.** *DPM-SNC using a 1-WL-GNN is as powerful as the AGG-WL test in distinguishing non-isomorphic graphs.*

---

[5] The node labels $y$ is replaced to the graph color $\chi$, and applying a latent variable is interpreted as applying an augmentation to the given graph.

*Proof.* We show that if two graphs $G, H$ are distinguishable by AGG-WL test, it is also distinguishable by the DPM-SNC. Assume that the two graphs $G, H$ are distinguishable by the AGG-WL. Then, the following condition is satisfied.

$$\chi_G^{\text{AGG}} = \{\!\!\{\chi_{G^m}^T : m \in [|\mathcal{Z}^G|]\}\!\!\} \neq \chi_H^{\text{AGG}} = \{\!\!\{\chi_{H^m}^T : m \in [|\mathcal{Z}^H|]\}\!\!\},$$

where the AGG-WL produces distinct sets of refined graph colors for $G, H$. Next, we discuss how the DPM-SNC can distinguish $G, H$. To this end, we first assume a GNN which is as powerful as the 1-WL test [20], denoted as 1-WL-GNN. Specifically, we denote $\chi_G^{\text{GNN}}$ as a graph color refined by the 1-WL-GNN, where $\chi_{G^m}^T \neq \chi_{H^n}^T$ implies $\chi_{G^m}^{\text{GNN}} \neq \chi_{H^n}^{\text{GNN}}$ for any augmented graph pair $G^m, H^n$. Then, under the $\chi_G^{\text{AGG}} \neq \chi_H^{\text{AGG}}$, the following condition is also satisfied.

$$\{\!\!\{\chi_{G^m}^{\text{GNN}} : m \in [|\mathcal{Z}^G|]\}\!\!\} \neq \{\!\!\{\chi_{H^m}^{\text{GNN}} : m \in [|\mathcal{Z}^H|]\}\!\!\}.$$

Here, we assume that DPM-SNC utilizes this 1-WL-GNN to output a graph color conditioned on an augmented graph, i.e., $p_\theta(\chi|G, z)$. Then, we can connect $p_\theta(\chi|G, z^m)$ with $\chi_{G^m}^{\text{GNN}}$ as follows:

$$p_\theta(\chi|G, z^m) = \mathbb{I}[\chi = \chi_{G^m}^{\text{GNN}}].$$

where $\mathbb{I}[a = b]$ is an indicator function whose value is 1 if $a = b$ and 0 otherwise. Next, we also assume that $p(z|G)$ is a uniform distribution over $\mathcal{Z}^G$. Then one can show that:

$$p_\theta(\chi|G) = \frac{1}{|\mathcal{Z}^G|} \sum_{m \in [|\mathcal{Z}^G|]} \mathbb{I}[\chi = \chi_{G^m}^{\text{GNN}}].$$

Then it follows that $p_\theta(\chi|G) \neq p_\theta(\chi|H)$ since $\{\!\!\{\chi_{G^m}^{\text{GNN}} : m \in [|\mathcal{Z}^G|]\}\!\!\} \neq \{\!\!\{\chi_{H^m}^{\text{GNN}} : m \in [|\mathcal{Z}^H|]\}\!\!\}$. Therefore, $G, H$ are distinguishable by DPM-SNC. □

Our proof is valid when considering the multiple outputs. However, our practical implementation of DPM-SNC does not use multiple outputs at inference time since the DPM only considers the multiple random variables in the training objective.[6] To this end, in Appendix F, we also investigate the inference scheme which aggregates multiple outputs to make final predictions.[7]

---

[6]Our inference method is described in Appendix C.1

[7]In practice, we also use a Gaussian diffusion for defining $p(z|G)$, which still allows the GNN to consider infinite augmentations.

## C  Implementation

In this section, we provide more details on how we implement the DPM-SNC for experiments.

### C.1  Transductive settings

Here, we provide the detailed implementation of DPM-SNC for transductive node classification.

**Model architecture.** We parameterize the residual function $\epsilon_{\boldsymbol{\theta}}(\boldsymbol{y}^{(t)}, G, t)$ of reverse diffusion step using a $L$-layer message-passing GNN as follows:

$$\epsilon_{\boldsymbol{\theta}}(\boldsymbol{y}^{(t)}, G, t) = g(\boldsymbol{h}^{(L)}),$$
$$h_i^{(\ell)} = (\text{COMBINE}^{(\ell)}(h_i^{(\ell-1)}, a_i^{(\ell)}) + f(t)) \| y_i^{(t)}),$$
$$a_i^{(\ell)} = \text{AGGREGATE}^{(\ell)}(\{h_j^{(\ell-1)} | (i, j) \in \mathcal{E}\}),$$

where $g(h^{(L)})$ is a multi-layer perceptron that estimates the residual using the final node represen-tation. AGGREGATE$(\cdot)$ and COMBINE$(\cdot)$ functions are identical to the backbone GNN, and $\cdot \| \cdot$ indicates the concatenation. Here, $h_i^0$ is $x_i \| y_i^{(t)}$. The $f(\cdot)$ is a sinusoidal positional embedding function [62]. We fix the dimension of sinusoidal positional embedding to 128.

**Buffer construction.** Following the temperature annealing approach of Qu et al. [7] in sampling pseudo-labels for optimization, we also control the temperature of randomness in obtaining $\boldsymbol{y}_U$ from $p_{\boldsymbol{\theta}}(\boldsymbol{y}_U | G, \boldsymbol{y}_L)$ for buffer construction. To be specific, we use the variance multiplied by the temperature $\tau \in [0, 1]$, instead of the original variance in the reverse diffusion step, e.g., setting $\tau$ to zero makes the deterministic sampling.

**Inference.** To make final predictions $\boldsymbol{y}_U$ from $p_{\boldsymbol{\theta}}(\boldsymbol{y}_U | G, \boldsymbol{y}_L)$ for evaluation, we eliminate the randomness in inference time, i.e., set temperature $\tau$ to zero.[8] If the target is one-hot relaxation of discrete labels, we also discretize the final prediction by choosing a dimension with maximum value.

### C.2  Inductive setting

Here, we provide the details of DPM-SNC for inductive node classification and graph algorithmic reasoning. For the inductive node classification, we use the same model architecture as in the transductive setting and use the deterministic inference strategy. For the graph algorithmic, we modify DPM-SNC to perform edge-wise prediction.

**Graph algorithmic reasoning.** Since the targets of the graph algorithmic reasoning task are defined on the edge-level, we apply a diffusion process to the edge labels. We then recover the edge-level noisy labels through the reverse process.

The denoising model architecture in the reverse process has a similar architecture to the model architecture of IREM [26]. Specifically, the noisy edge target $\boldsymbol{y}^{(t)}$ is updated as follows. First, the noisy edge labels $\boldsymbol{y}^{(t)}$ and edge features are concatenated and passed through to the GNN layer, which aggregates them to obtain the node representation. Next, we apply element-wise addition of the time embedding vector to the node representation. Then, we concatenate a pair of node representations and noisy targets for the given edges and then apply a two-layer MLP to update edge labels.

In contrast to the node classification, we maintain randomness at inference time, i.e., we use the stochastic reverse process for obtaining edge labels. This approach is consistent with the IREM, which also includes randomness at inference time.

---

[8]We also investigate various stochastic inference strategies in Appendix F

# D Data statistics

## D.1 Synthetic data

We generate $1000 \times 2$ non-attributed cyclic grid and $100 \times 2$ non-attributed cyclic grid for scattered and localized training nodes scenarios, respectively. Then, we split 30%, 30%, and 40% of the entire nodes into training, validation, and test nodes.

- *Scattered training nodes*: We randomly sample nodes in the graph to split them into training, validation, and test nodes.[9]
- *Localized training nodes*: We select the nodes in the region within the $30 \times 2$ grid as training nodes. Then, we randomly sample the remaining nodes in the graph to split them into validation and test nodes.

For illustrative purposes, we also describe both scenarios in Figure 10 with smaller graphs.

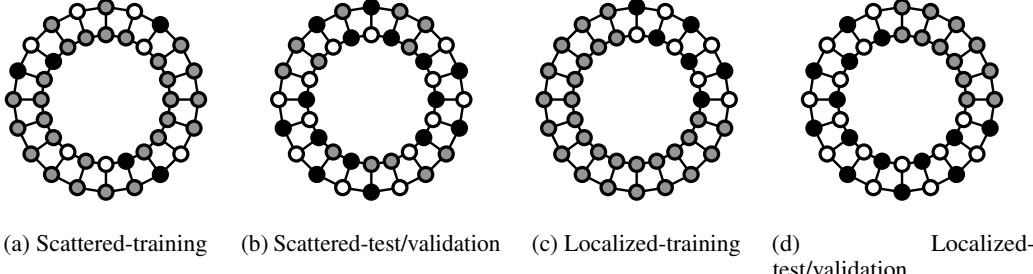

(a) Scattered-training    (b) Scattered-test/validation    (c) Localized-training    (d) Localized-test/validation

Figure 10: Illustration of two scenarios. The non-gray nodes represent nodes in each split.

## D.2 Transductive node classification datasets

Table 6: The data statistics of transductive node classification datasets.

| Dataset | ♯ nodes | ♯ edges | ♯ features | ♯ classes | ♯ (training/validation/test) nodes |
|---|---|---|---|---|---|
| Pubmed [22] | 19717 | 44338 | 500 | 3 | (60/500/1000) |
| Cora [22] | 2708 | 5429 | 1433 | 7 | (140/500/1000) |
| Citeseer [22] | 3327 | 4732 | 3703 | 6 | (120/500/1000) |
| Photo [23] | 7487 | 119043 | 745 | 8 | (160/240/7087) |
| Computers [23] | 13381 | 34493 | 767 | 10 | (200/300/12881) |
| Roman [24] | 22662 | 32927 | 300 | 18 | (11331/5665/5666) |
| Ratings [24] | 24492 | 93050 | 300 | 5 | (12246/6123/6123) |

Here, we consider a graph with partially labeled nodes. We describe the data statistics in Table 6.

## D.3 Inductive node classification datasets

Table 7: The data statistics of inductive node classification datasets.

| Dataset | ♯ features | ♯ classes | (training/validation/test) data | | |
|---|---|---|---|---|---|
| | | | ♯ graphs | Avg. ♯ nodes | Avg. ♯ edges |
| Pubmed [9] | 500 | 3 | (60/500/1000) | (6.0/5.4/5.6) | (6.7/5.8/6.7) |
| Cora [9] | 1433 | 7 | (140/500/1000) | (5.6/4.9/4.7) | (7.0/5.8/5.3) |
| Citeseer [9] | 3703 | 6 | (120/500/1000) | (4.0/3.8/3.8) | (4.3/4.0, 3/8) |
| PPI [25] | 500 | 3 | (20/2/2) | (2245.3/3257.0/2762.0) | (61318.4/99460.0/80988.0) |

Here, we consider datasets consists of a set of graphs. We describe the data statistics in Table 7.

---

[9]Additionally, we also consider training with an additional $2 \times 20$ cyclic grid for visualization in Figure 1.

## D.4 Graph algorithmic reasoning datasets

Following Du et al. [26], we generate training graphs in each training step. Here, the training graphs are composed of graphs of varying sizes, ranging from two to ten nodes. The node features are initialized to zero, and the labels are defined on the edges, e.g., the shortest distance between two nodes. Then, we evaluate performance on graphs with ten nodes. Furthermore, we also use graphs with 15 nodes to evaluate generalization capabilities.

# E   Experiments setup

In this section, we describe the detailed experimental setup. For all experiments, we use a single GPU of NVIDIA GeForce RTX 3090. The hyper-parameters for each experiment are described in the following subsections.

## E.1   Synthetic dataset

In this experiment, we implement each method with a one-layer GCN with 16 hidden dimensions. We search the learning rate within $\{1e{-}3, 5e{-}3, 1e{-}2\}$ for all methods. Other hyper-parameters of each method follow their default settings. For DPM-SNC, we fix the diffusion step to 100. We also set the size of the buffer to 50 and insert five samples into the buffer for every 30 training step. We use a pre-trained mean-field GNN until the buffer is updated 10 times.

## E.2   Transductive node classification

Table 8: The hyper-parameter search ranges for the homophilic graph. For hyper-parameters without a specific method in parentheses, it applies to all methods in the respective category.

| Method | Hyper-parameters | Search range |
|---|---|---|
| All methods | learning rate
weight decay | $\{1e{-}3, 5e{-}3, 1e{-}2\}$
$\{1e{-}3, 5e{-}3, 1e{-}2\}$ |
| GNN-based methods (LPA, GMNN, G$^3$NN, CLGNN, DPM-SNC) | number of layers
hidden dimension
weight of constraints for structured-prediction (LPA, G$^3$NN)
pseudo-labels sampling temperature (GMNN, CLGNN, DPM-SNC) | $\{2, 4\}$
$\{64, 128\}$
$\{0.1, 1.0, 10.0\}$

$\{0.1, 0.3, 1.0\}$ |
| Non-GNN methods (LP, PTA) | number of label propagation
hidden dimension (PTA)
damping factor | $\{10, 100\}$
$\{64, 128\}$
$\{0.1, 0.3, 0.5\}$ |

**Homophilc graph.** We describe the hyper-parameter search ranges in Table 8. Additionally, we apply dropout with $p = 0.5$ except for LP. Other hyper-parameters of each method follow their default settings. For DPM-SNC, we fix the diffusion step to 80. We also set the size of the buffer to 50 and insert five samples into the buffer for every 100 training step. We use a pre-trained mean-field GNN until the buffer is updated 20 times.

**Heterophilic graph.** Here, we describe the hyper-parameter settings for DPM-SNC as we use the numbers reported by Plantov et al. [24] for baselines. We describe the hyper-parameter search ranges in Table 9. Additionally, we apply dropout with $p = 0.5$, and we fix the diffusion step to 80. We also set the size of the buffer to 50 and insert five samples into the buffer for every 100 training step. We use a pre-trained mean-field GNN until the buffer is updated 100 times.

Table 9: The hyper-parameter search ranges of DPM-SNC for the heterophilic graph.

| Hyper-parameters | Search range |
|---|---|
| learning rate | $\{3e{-}5, 1e{-}4, 3e{-}4\}$ |
| weight decay | $\{0, 1e{-}5, 1e{-}4\}$ |
| number of layers | $\{2, 4\}$ |
| hidden dimension | $\{256, 512\}$ |
| sampling temperature | $\{0.1, 0.3, 1.0\}$ |

## E.3   Inductive node classification

Table 10: The hyper-parameter search ranges of all methods for the inductive node classificaiton.

| Hyper-parameters | Search range |
|---|---|
| learning rate | $\{1e{-}3, 5e{-}3, 1e{-}2\}$ for small-scale graphs and $\{3e{-}5, 1e{-}4, 3e{-}4\}$ for huge-scale graphs |
| weight decay | $\{1e{-}3, 5e{-}3, 1e{-}2\}$ for small-scale graphs and $\{0, 1e{-}5, 1e{-}4\}$ for huge-scale graphs |
| number of layers | $\{2, 4\}$ |
| hidden dimension | $\{64, 128\}$ for small-scale graphs and $\{512, 1024\}$ for huge-scale graphs |

We describe the hyper-parameter search ranges in Table 10. For the small-scale graph datasets, i.e., Pubmed, Cora, and Citeseer, we apply dropout with $p = 0.5$. For the huge-scale graph datasets, i.e., PPI, we include the linear skip connection between each GNN layer. Other hyper-parameters of each method follow their default settings. For DPM-SNC, we fix the diffusion step to 80.

### E.4 Algorithmic reasoning

Here, we describe the hyper-parameter settings for DPM-SNC as we use the numbers reported by Du et al. [26] for baselines. We search the learning rate and weight decay within $\{1e-4, 3e-4, 1e-3\}$ and $\{0, 1e-5, 1e-4\}$, respectively. The hyper-parameters of the model are the same as the model implementation of IREM, using a three-layer GINEConv [63] with a $128$ hidden dimension. We fix the diffusion step to $80$.

# F    Additional experiments

**Ablations on various inference schemes.**  We also study various inference strategies for our DPM-SNC. We first investigate how temperature control affects label inference in real-world node classification tasks. In Figure 11(a), we plot the changes in accuracy for various temperatures $\tau$. One can see that reducing the randomness of DPM-SNC gives a better prediction in real-world node classification tasks.

We also consider sampling various numbers of predictions for node-wise aggregation to improve performance. In Figure 11(b), even the number of samples is increased to $2^{10}$, the deterministic and the node-wise aggregation inference schemes show similar accuracy, and there are only minor performance improvements on Citeseer. In practice, we use deterministic inference in the node classification tasks since the node-wise aggregation requires a relatively long time.

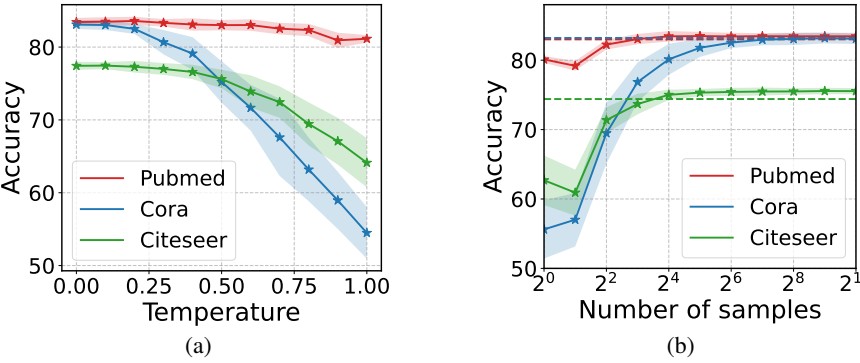

Figure 11: (a) Accuracy with varying temperature. (b) Accuracy with the varying number of samples. The dashed line represents the accuracy of the deterministic inference scheme.

