# OpenReview forum: "Diffusion Probabilistic Models for Structured Node Classification"
_NeurIPS.cc/2023/Conference — NeurIPS 2023 poster_

### Official Review · Reviewer_w4tc · 2023-07-05

**Soundness:** 3 good
**Presentation:** 3 good
**Contribution:** 3 good
**Rating:** 5
**Confidence:** 3

**Summary:**

This paper focuses on structured node classification on graphs by integrating diffusion probabilistic models to generate label distributions for unknown nodes. The authors design a novel training algorithm to apply DPMs, and a new variational lower bound. Theoretical analysis proves the proposed model is strictly more powerful than the classic 1-WL test. Experimental results also show its superiority over baseline methods on both inductive and transductive settings.

**Strengths:**

- The paper is well written and structured.
- Application of diffusion problistic models in node classification is novel.
- The theoredical analysis of 1-WL test is convincing.

**Weaknesses:**

- The use of diffusion problistic model is not well motivated? The reason for using diffusion process for partially labeled graphs is not explained.
- It lacks a analysis of computational complexity of DPM-SNC, since it requires multiple steps of sampling.

**Questions:**

- Please explain the reason of introducing diffusion problistic models. How can diffusion process improves GNNs on partially labeled graphs?
- What is the computational complexity of DPM-SNC?

**Limitations:**

Please refer to the weaknesses.

---

> ### Author Rebuttal · Authors · 2023-08-09
>
> Dear reviewer w4tc,
>
> We express our deep appreciation for your time and insightful comments. In what follows, we address your comments one by one.
>
> ---
>
> **W1/Q1. The diffusion problistic model is not well motivated. Why use diffusion process for partially labeled graphs?**
>
> For the partially labeled graphs, we use the diffusion probabilistic model to incorporate known label information into the unknown label predictions. Specifically, known labels can be benefical by providing *an additional degree of freedom to incorporate dependencies between labels* (**Section 3.1** and **Section 3.2**), e.g., Figure 1. Among the structured node classification baselines, reverse diffusion is especially powerful since it enables *non-linear updates* (via GNN) to incorporate label dependencies *while easily conditioning on the known labels* (via manifold-based sampling [1]).
>
> We also note how this motivation is reinforced by the state-of-the-art performance of the diffusion probabilistic model for the well-studied image inpainting problem: predicting unknown pixels from known pixels. This is a similar task to incorporate known (pixel) labels to predict unknown (pixel) labels.
>
> ---
>
> **W2/Q2. It lacks a analysis of computational complexity of DPM-SNC, since it requires multiple steps of sampling.**
>
> The time complexity is the number of diffusion steps $T$ times the time complexity of the backbone GNN that parameterizes the reverse diffusion. The space complexity is identical to the corresponding GNN, as reverse diffusion step $t$ only requires results obtained from the previous step $t+1$. For example, consider parameterizing DPM with GCN, where we use a hidden dimension of $F$ and $L$ layers. Since the GCN has $O(L|\mathcal{V}|F^2+L|\mathcal{E}|F)$ time complexity, the reverse process requires $O(T(L|\mathcal{V}|F^2+L|\mathcal{E}|F))$ time complexity.
>
> ---
>
> **References**
>
> [1] Chung et al., Improving Diffusion Models for Inverse Problems using Manifold Constraints, NeurIPS 2022
>
> [2] You et al., L$^2$-GCN: Layer-Wise and Learned Efficient Training of Graph Convolutional Networks, CVPR 2020

---

### Official Review · Reviewer_7A8R · 2023-07-07

**Soundness:** 3 good
**Presentation:** 3 good
**Contribution:** 3 good
**Rating:** 7
**Confidence:** 4

**Summary:**

The paper addresses the issue of structured node classification on graphs that consider dependencies between node labels. While GNNs have shown great success in machine learning on graphs, they predict node labels independently.
The authors propose a novel framework, DPM-SNC (Diffusion Probabilistic Models for Structured Node Classification), in which they use DPMs to learn a joint label distribution over both known and unknown labels while making predictions based on known labels using manifold-constrained sampling. Additionally, they present a novel training algorithm for DPMs. Theoretical analysis shows that it enhances the expressive power of GNNs. Experiments across multiple settings demonstrate the effectiveness of the proposed approach.


**Strengths:**

**Originality**

This paper ...
- combined GNNs and DPMs for structured node classification.
- proposed using manifold-constraint sampling for the transductive setting.
- gave a novel expressivity metric, AGG-WL and proved that DPM-SNC is more powerful than its backbone 1-WL-GNN.

**Quality**

In general, the math is solid and the claims are intuitive and/or empirically validated. The experiments are extensive.

**Clarity**

The paper is well-written and easy to follow.

**Significance**

The paper makes interesting and inspiring theoretical connections. Realistically, I am not so sure that this technique would be adopted in a production setting considering its cost and complexity.

**Weaknesses:**

No major flaws. See questions.

**Questions:**

1. What are the training and inference costs of the proposed method? How do they compare to the baselines?
2. Regarding model training, do you use the same objective as DDPM to optimize the GNN parameters? Based on my understanding of your paper and code (correct me if I am wrong), the $q(y_U|y_L)$ term in the objective is not used to update $\theta$. Instead, it is optimized by updating the buffer $\mathcal{B}$. Maybe the "Training algorithm" paragraph could be revised so that it better reflects the proposed method?
3. Among all kinds of generative models, why do you choose DPMs?

**Limitations:**

I do not see any significant, unreported negative societal impact.

---

> ### Author Rebuttal · Authors · 2023-08-09
>
> Dear reviewer 7A8R,
>
> We express our deep appreciation for your time and insightful comments. In what follows, we address your comments one by one.
>
> ---
>
> **Q1. What are the training and inference costs of the proposed method? How do they compare to the baselines?**
>
> To resolve your question, we provide the statistics on training and inference time in **Table F**.
>
> *Table F. Time costs (sec) for (training/inference).*
> \begin{array}{lccc}
> \hline
> \text{Methods}  & \text{Pubmed} & \text{Cora} & \text{Citeseer} \newline
> \hline
> \text{GCN} & 1.96/0.004 & 2.89/0.003 & 2.09/0.004
> \newline
> \text{GCN+CLGNN} & 639.04/0.074 & 608.19/0.061 & 862.40/0.078
> \newline
> \text{GCN+DPM-SNC} & 362.75/0.362 & 265.73/0.335 & 464.52/0.397
> \newline
> \hline
> \end{array}
>
> Overall, the structured prediction requires long training time to train the complicated node-wise label dependencies compared to learning only independent node-wise predictions. For prolonged inference time, we believe accelerating diffusion models, e.g., denoising diffusion implicit models (DDIM) [1], is an important future research direction (as described in limitations).
>
> ---
>
> **Q2. Maybe the "Training algorithm" paragraph could be revised so that it better reflects the original DPM objective and updating buffer instead of the variational distribution.**
>
> Indeed, we use the original DPM objective given samples from the variational distribution. We will modify the paragraph to better reflect this point. However, it is worth noting that updating the buffer is equal to updating the variational distribution represented by the empirical distribution of the buffer, as mentioned in Line 168 and 176.
>
> ---
>
> **Q3. Among all kinds of generative models, why do you choose DPMs?**
>
> We chose DPM due to its ability to incorporate known labels easily as an additional condition. Once the generative distribution is trained, considering additional conditions, i.e., known labels, becomes non-trivial for other generative models like variational autoencoders. However, DPM defines a generative distribution using gradients of log density across multiple time steps, which enables incorporating known labels through manifold-based sampling [2].
>
> ---
>
> **References**
>
> [1] Song et al., Denoising Diffusion Implicit Models, ICLR 2021
>
> [2] Chung et al., Improving Diffusion Models for Inverse Problems using Manifold Constraints, NeurIPS 2022

---

> > ### Comment · Reviewer_7A8R · 2023-08-14
> > **Reply to the authors**
> >
> > Thank you for the detailed clarifications. I have no further comment.

---

> > > ### Author Response · Authors · 2023-08-16
> > >
> > > Dear reviewer 7A8R,
> > >
> > > We are happy to hear that our rebuttal adequately addressed your questions! We also appreciate your insightful comments on our works.

---

### Official Review · Reviewer_JeF8 · 2023-07-07

**Soundness:** 3 good
**Presentation:** 3 good
**Contribution:** 2 fair
**Rating:** 6
**Confidence:** 4

**Summary:**

This paper proposes a diffusion probability model learning method for structured node classification, named DPM-SNC. The contributions of this paper are as follows. First, this paper leverages the high capacity of DPMs in learning joint label dependency and predicting posterior distribution conditioned on partially known data by devising DPM-SNC. Second, this paper introduces a new variational lower bound to train a DPM on partially labelled graphs. Moreover, this paper proposes AGG-WL test to theoretically analyse the expressive power of DPM-SNC using 1-WL-GNNs, which is more powerful than 1-WL-GNNs. Finally, detailed experiments on both transductive and inductive settings demonstrate DPM-SNC’s impressive performance.

**Strengths:**

There are strengths of proposed method:
1.	Applying DPM to learning partially labeled node label, to the best of our knowledge, is novel.

2.	The experiments are detailed analyzed with both synthetic and real world data, which points out the strength of the method in different scenarios. The algorithmic reasoning shows the influence of different algorithmic elements in DPM, with great performance compared with baseline.


**Weaknesses:**

However, there are several drawbacks:
1.	The idea to apply DPM to learn the joint distribution over the labels is novel. However, this paper has little novelty to DPM itself, which utilizes manifold-constrained sampling from previous works.
2.	The analysis for expressive power follows the color assignments description in 1-WL test. The details of AGG-WL, the perturbations, and the aggregation need more explanation.


**Questions:**

1.	Add more explanation to AGG-WL.

**Limitations:**

Authors have adequately addressed the limitations in this paper.

---

> ### Author Rebuttal · Authors · 2023-08-09
>
> Dear reviewer JeF8,
>
> We express our deep appreciation for your time and insightful comments. In what follows, we address your comments one by one.
>
>
> ---
>
> **W1. What is the novelty of this work compared to DPM itself?**
>
>
> When compared to the exisiting works on DPM, our work is the first to (a) train DPM in a semi-supervised setting using a new variational lower bound and (b) theoretically analyze the expressive power of a DPM by deriving an analog based on the Weisfeiler-Lehman (WL) test, i.e., aggregated WL test. We believe such contributions are quite novel even in the entire research field of DPMs.
>
> ---
>
> **W2/Q1. The details of AGG-WL, the perturbations, and the aggregation need more explanation.**
>
> To alleviate your concern, we will add a more detailed description of AGG-WL in Section 5. As stated in **Appendix B.2**, AGG-WL applies perturbations into the initial graph to create multiple augmented graphs, and aggregates the results of the 1-WL algorithm on each augmented graph. The perturbations are node-wise concatenations of node attributes $\boldsymbol{x}$ with binary random features $\boldsymbol{z}$. The aggregation represents hashing a multiset of the augmented graph colors to obtain the final graph color.
>
> If you additionally point out any unclear aspects, we would be happy to provide a detailed description.

---

> > ### Comment · Reviewer_JeF8 · 2023-08-18
> >
> > Thanks for author's response. And I'd like to keep my score.

---

### Official Review · Reviewer_kJkj · 2023-07-07

**Soundness:** 3 good
**Presentation:** 3 good
**Contribution:** 3 good
**Rating:** 5
**Confidence:** 3

**Summary:**

This paper investigates the use of diffusion probabilistic models (DPM) to model node-wise label dependencies for structured node classification. The proposed DPM-SNC generalizes diffusion probabilistic models to work on partially labeled graphs in the transduction settings. The proposed model is proved to be more expressive than the classic 1-WL test. Experimental results demonstrate the effectiveness of the proposed model.

**Strengths:**

1. This paper leverages the merits of diffusion probabilistic models (DPM) for structured node classification. A new training algorithm is proposed, which generalises DPM to work on partially labeled graphs.
2. The theoretical analysis proves the improved expressive power of the proposed model.
3. The design of experiments is thorough and the reported results demonstrate the effectiveness of the proposed model on various graph-based tasks.

**Weaknesses:**

1. The idea of using diffusion models is borrowed from the field of computer vision, where the diffusion and reverse process is more intuitive. However, why structured node classification would benefit from the diffusion and reverse process? Why the reverse process can be defined as a Gaussian distribution on graphs?

2. It is unclear how the proposed model is applied to inductive node classification and what benefits the diffusion model can offer in inductive settings.

3. As observed from Table 4, the performance of the proposed method is comparable or with limited improvements. More elaborations on these points are necessary. Also, the datasets used for inductive settings (except for PPI) are mostly used for transductive settings. Experiments on other relevant datasets should be added for inductive settings.



**Questions:**

See weaknesses above.

**Limitations:**

No potential negative societal impact of this work.

---

> ### Author Rebuttal · Authors · 2023-08-09
>
> Dear reviewer kJkj,
>
> We express our deep appreciation for your time and insightful comments. In what follows, we address your comments one by one.
>
> ---
>
> **W1-1. How does reverse diffusion benefit structured node classification?**
>
> The reverse diffusion outputs a predictive distribution for structured node classification with *an additional degree of freedom to incorporate dependencies between labels* (**Section 3.1** and **Section 3.2**). This is particularly useful for prediction on partially labeled graphs, where the known labels provide additional information about the unknown labels, e.g., **Figure 1**. Among the structured node classification baselines, reverse diffusion is especially powerful since it enables *non-linear updates* (via GNN) to incorporate label dependencies *while easily conditioning on the known labels* (via manifold-based sampling [1]).
>
> As a more intuitive analogy, our benefit is akin to using diffusion models for image inpainting, i.e., filling out the missing pixels in an incomplete image. Similar to how diffusion models are suitable for predicting unknown pixels from known ones, diffusion model is also suitable for predicting unknown node labels from the known ones.
>
> ---
>
> **W1-2. How can the reverse process be defined as a Gaussian distribution on graphs?**
>
> To define the diffusion and reverse process for node labels, we relax the node label into a continuous one-hot vector and apply Gaussian diffusion to this vector (similar to applying Gaussian diffusion to an RGB vector in a pixel).
>
> ---
>
> **W2. How is the model applied to inductive node classification and what benefits the diffusion model can offer?**
>
> We train our model with the same DPM-SNC algorithm. However, the set of unlabeled nodes $y_{U}$ is empty during training in the inductive setting, so our algorithm skips manifold-constrained sampling and buffer updates. At evaluation, we infer the labels without any conditioning on the known labels.
>
> Next, using a diffusion model benefits inductive node classification through propagating information via label-wise dependencies, similar to other inductive neural structured prediction baselines [2]. We also note how the diffusion model enhances the expressive power of GNNs (**Section 5**) even for inductive node classification.
>
> ---
>
> **W3-1. It is necessary to elaborate on why the improvement is limited for inductive node classification (Table 4).**
>
> The improvements are relatively small since there are no known labels, which is a significant information to be incorporated as inputs for structured prediction. Nevertheless, we would like to point out how our framework improves the performance over baselines, hence useful.
>
> ---
>
> **W3-2. Except for PPI, the datasets used for inductive settings are mostly used for transductive settings.**
>
> We first clarify that our experiments on algorithmic reasoning (**Table 5**) are also inductive.
>
> *Table D. Graph algorithmic reasoning tasks on graphs with 10/15 nodes.*
> \begin{array}{lccc}
> \hline
> \text{Methods}  & \text{Edge copy} & \text{Connected components} & \text{Shortest path} \newline
> \hline
> \text{Feedforward} & 0.3016/0.3124 & 0.1796/0.3460 & 0.1233/1.4089
> \newline
> \text{IREM} & 0.0019/\textbf{0.0019} & 0.1424/0.2171 & 0.0274/0.0464
> \newline
> \text{DPM-SNC} & \textbf{0.0011}/0.0038 & \textbf{0.0724}/\textbf{0.1884} & \textbf{0.0138}/\textbf{0.0286}
> \newline
> \hline
> \end{array}
>
> To further incorporate your concern, we experiment on additional datasets (PPI-{1,2,10} [2], DHFR [3]) in **Table E**. One can observe how our method still demonstrates a consistent improvement over the baselines.
>
> *Table E. Inductive node classification performance (F1 score for PPI and accuracy for DHFR).*
> \begin{array}{lccc}
> \hline
> \text{Methods}  & \text{PPI-1} & \text{PPI-2} & \text{PPI-10} & \text{DHFR} \newline
> \hline
> \text{GCN} & 65.08 \pm{0.39} & 71.67 \pm{0.22} & 96.16 \pm{0.03} & 89.12 \pm{0.71} &
> \newline
> \text{GCN+CLGNN} & 66.02 \pm{0.42} & 71.82 \pm{0.33} & 96.28 \pm{0.04} & 91.48 \pm{1.06} &
> \newline
> \text{GCN+SPN} & 68.12 \pm{0.19} & 74.05 \pm{0.15} & 95.34 \pm{0.14}  & 89.95 \pm{0.78} &
> \newline
> \text{GCN+DPM-SNC} & \mathbf{70.08} \pm{0.43} & \mathbf{76.20} \pm{0.43} & \mathbf{97.00} \pm{0.09} & \mathbf{92.26} \pm{0.63} &
> \newline
> \hline
> \end{array}
>
> ---
>
> **References**
>
> [1] Chung et al., Improving Diffusion Models for Inverse Problems using Manifold Constraints, NeurIPS 2022
>
> [2] Qu et al., Neural Structured Prediction for Inductive Node Classification, ICLR 2022
>
> [3] Wen et al., Meta-Inductive Node Classification across Graphs, SIGIR 2021

---

### Official Review · Reviewer_rLqd · 2023-07-07

**Soundness:** 4 excellent
**Presentation:** 3 good
**Contribution:** 3 good
**Rating:** 6
**Confidence:** 2

**Summary:**

This paper proposes a new framework called the Diffusion Probabilistic Model for Structured Node Classification (DPM-SNC), which leverages diffusion probabilistic models for structured node classification on graphs. The framework aims to account for dependencies between node labels. The two key aspects of DPM-SNC are: (a) learning a joint distribution over the labels using an expressive reverse diffusion process and (b) making predictions based on known labels using manifold-constrained sampling. The authors test DPM-SNC in diverse scenarios, including partially labeled graphs in a transductive setting, as well as unlabeled graphs and in an inductive setting.

**Strengths:**

1. The paper effectively applies probabilistic diffusion, one of the hottest techniques in the community, to the structured node classification problem.
2. DPM-SNC can be used in combination with any graph neural network (GNN) and to enhance performance on node classification tasks. This improvement is not solely empirical, as the authors provide theoretical analysis tying it to enhanced theoretical expressive power.
3. The authors conduct a comprehensive experimental evaluation, showing DPM-SNC's superior performance in both transductive and inductive node classification tasks.

**Weaknesses:**

1. Although it is understandable that certain technical details are relegated to the appendix due to space limitations, the main paper could benefit from being more self-contained. This could be achieved by shortening the introduction or reducing some figures. Without referring to the appendix, especially for readers without expertise in diffusion models, it's difficult to grasp exactly how DPM-SNC makes predictions.
2. The paper lacks a discussion or comparison with similar works that combine deep neural networks and label propagation for structured node classification [1, 2]. These works could be used as additional baselines or in discussions related to the proposed method.
3. It appears that a limitation of the proposed method is that the diffusion process operates on (noisy versions of) labels of nodes, not their representation vectors. As a result, while DPM-SNC seems to work well for node classification problems, it's unclear whether the trained model could be effectively used as a general node embedding model for solving edge- or graph-level problems, or multi-label classification problems.

[1] Huang et al., Combining Label Propagation and Simple Models Outperforms Graph Neural Networks, ICLR 2021
[2] Yoo et al., Belief Propagation Network for Hard Inductive Semi-Supervised Learning, IJCAI 2019

**Questions:**

1. Why did the authors exclusively use GCNII for the proposed method and not for the competitor models?

**Limitations:**

The authors have acknowledged the limitations of their approach.

---

> ### Author Rebuttal · Authors · 2023-08-09
>
> Dear reviewer rLqd,
>
> We express our deep appreciation for your time and insightful comments. In what follows, we address your comments one by one.
>
> ---
>
> **W1. For readers without expertise in diffusion models, the main paper could benefit from being more self-contained.**
>
> We will follow your suggestion and make space to introduce diffusion models in **Section 3.2**, using contents from **Appendix A** and **Appendix C**. We will also provide more details on the implementation, e.g., we relax the discrete labels to one-hot vectors to apply Gaussian forward diffusion process.
>
> ---
>
> **W2. Two similar works could be used in discussion or as additional baselines.**
>
> We will incorporate your valuable suggestion in our future manuscript. Both works are based on label propagation that linearly updates the node-wise predictions based on neighboring predictions. DPM-SNC mainly differs by using non-linear updates. In **Table A**, we show how our method outperforms the new baselines.
>
> *Table A. Transductive node classification performance.*
> \begin{array}{lccc}
> \hline
> \text{Methods}  & \text{Pubmed} & \text{Cora} & \text{Citeseer} \newline
> \hline
> \text{C} \\& \text{S} & 77.3 \pm{0.0} & 80.2\pm{0.0} & 69.5\pm{0.0}
> \newline
> \text{BPN} & 78.2 \pm{1.5}& 82.5 \pm{0.9} & 73.3 \pm{0.7}
> \newline
> \text{GCN+DPM-SNC} & \textbf{83.0} \pm{0.9} & \textbf{83.2} \pm{0.5} & \textbf{74.4} \pm{0.5}
> \newline
> \hline
> \end{array}
>
> ---
>
> **W3. Can your work effectively solve edge-level, graph-level, or multi-label classification?**
>
> Yes. We already have solved edge-level tasks (algorithmic reasoning) and multi-label classification (PPI) in **Table 5** and **Table 4**, respectively. For graph-level problems, we did not conduct the experiments since our main focus is to use DPM-SNC for considering dependencies between multiple labels.
>
> Nevertheless, to further alleviate your concern, we additionally consider graph classification by assigning a graph label as identical node-wise labels. We construct the final graph-level prediction via mean pooling over node-wise predictions. In **Table B**, our method still improves base GNN in some graph-level problems (MUTAG and IMDB-B). We hypothesize that the performance is improved due to the (1) enhanced expressive power and (2) expanded receptive field of GNN with repeated reverse diffusion.
>
> *Table B. Graph classification performance.*
> \begin{array}{lccc}
> \hline
> \text{Methods}  & \text{MUTAG} & \text{IMDB-B} \newline
> \hline
> \text{GCN} & 86.1 \pm{5.5} & 73.2 \pm{5.6} &
> \newline
> \text{GCN+DPM-SNC} & \textbf{87.1} \pm{5.1} & \textbf{75.9} \pm{3.4}
> \newline
> \hline
> \end{array}
>
>
> ---
>
> **Q1. Why did the authors exclusively use GCNII for the proposed method and not for the competitor models?**
>
> We originally used GCNII to show our architecture-agnostic improvement of DPM-SNC on various GNNs. Nevertheless, to further resolve your concern, we conduct an additional experiment that combines GCNII with other competitors. In **Table C**, our method still outperforms other baselines.
>
>
> *Table C. Transductive node classification performance.*
> \begin{array}{lccc}
> \hline
> \text{Methods}  & \text{Pubmed} & \text{Cora} & \text{Citeseer} \newline
> \hline
> \text{GCNII} & 82.0\pm{0.8} & 84.0\pm{0.6} & 72.9\pm{0.5}
> \newline
> \text{GCNII+LPA} & 82.6\pm{2.1} & 82.8\pm{1.5} & 72.6\pm{0.4}
> \newline
> \text{GCNII+G3NN} & 83.3\pm{1.2} & 84.1\pm{0.9} & 73.8\pm{0.8}
> \newline
> \text{GCNII+GMNN} & 82.5\pm{1.0} & 83.3\pm{1.1} & 73.4\pm{0.3}
> \newline
> \text{GCNII+CLGNN} & 82.9\pm{0.9} & 84.3\pm{0.4} & 73.2\pm{0.6}
> \newline
> \text{GCNII+DPM-SNC} & \textbf{83.8}\pm{0.7}& \textbf{85.3}\pm{0.6} & \textbf{74.1}\pm{0.5}
> \newline
> \hline
> \end{array}

---

> > ### Comment · Reviewer_rLqd · 2023-08-14
> > **Thank you for the response**
> >
> > I appreciate the authors' clear and informative reply. I anticipate that these new experimental results will be replicated in the other two datasets, Photo and Computer.

---

> > > ### Author Response · Authors · 2023-08-16
> > >
> > > Dear reviewer rLqd,
> > >
> > > We are happy to hear that our rebuttal addressed your concerns! We will also incorporate additional experimental results on remaining datasets, i.e., Photo and Computer, based on **Table G**.
> > >
> > > *Table G. Transductive node classification performance.*
> > > \begin{array}{lcc}
> > > \hline
> > > \text{Methods}  & \text{Photo} & \text{Computer} \newline
> > > \hline
> > > \text{C}\\& \text{S} & 90.1 \pm{1.4}& 81.6 \pm{0.9}
> > > \newline
> > > \text{BPN} & 89.0\pm{1.0} & 80.4 \pm{1.4}
> > > \newline
> > > \text{GCN+DPM-SNC} & \textbf{92.2} \pm{0.8} & \textbf{84.1} \pm{1.3}
> > > \newline
> > > \hline
> > > \end{array}

---

### Decision · Program_Chairs · 2023-09-21

**Decision:**

Accept (poster)

**Comment:**

In this paper the authors address  structured node classification on graphs, where there are possible dependencies between the node labels.The authors develop a new approach that  leverages the diffusion probabilistic model for structured node classification (DPM-SNC). In addition to theoretical analysis, the authors present results of extensive evaluations showing DPM-SNC's superior performance compared to baselines. The reviews agreed that the paper presents a novel, technically solid and potentially impactful contribution. There were also some questions about novelty of the work with respect diffusion probabilistic models, training and inference costs, computational complexity, etc, which the authors addressed during the rebuttal phase.